

# The potential of satellite spectro-imagery for monitoring CO$_2$ emissions from large cities

Grégoire Broquet[1], François-Marie Bréon[1], Emmanuel Renault[1,*], Michael Buchwitz[2], Maximilian Reuter[2], Heinrich Bovensmann[2], Frédéric Chevallier[1], Lin Wu[1], Philippe Ciais[1]

[1]Laboratoire des Sciences du Climat et de l'Environnement, CEA-CNRS-UVSQ, UMR8212, IPSL, Gif-sur-Yvette, France
[2]Institute of Environmental Physics (IUP), University of Bremen FB1, Otto Hahn Allee 1, 28334 Bremen, Germany
*now at: Service hydrographique et océanographique de la marine, 29228 Brest, France

*Correspondence to:* Grégoire Broquet (gregoire.broquet@lsce.ipsl.fr)

**Abstract.** This study assesses the potential of 2 to 10 km resolution imagery of CO$_2$ concentrations retrieved from the Short Wave Infra Red measurements of a space borne passive spectrometer for monitoring the spatially integrated emissions from the Paris area. Such imagery could be provided by missions similar to CarbonSat, which was studied as a candidate Earth Explorer 8 mission by the European Space Agency (ESA). This assessment is based on Observing System Simulation Experiments (OSSEs) with an atmospheric inversion approach at city scale. The inversion system solves for hourly city CO$_2$ emissions and natural fluxes, or for these fluxes per main anthropogenic sector or ecosystem, during the 6 hours before a given satellite overpass. These 6 hours correspond to the period during which emissions produce CO$_2$ plumes that can be identified on the image from this overpass. The statistical framework of the inversion accounts for the existence of some prior knowledge about the hourly emissions from an inventory based on energy use and carbon fuel consumption statistics. The link between the hourly or sectorial emissions and the vertically-integrated column of CO$_2$ observed by the satellite is simulated using a coupled flux and atmospheric transport model. This coupled model is built with the information on the spatial and temporal distribution of emissions from the emission inventory produced by the local air-quality agency (Airparif) and a 2 km horizontal resolution atmospheric transport model. Tests are conducted for different realistic simulations of the spatial coverage, resolution, precision and accuracy of the imagery from sun-synchronous polar-orbing missions, corresponding to the specifications of CarbonSat and Sentinel-5 or extrapolated from these specifications. First, OSSEs are conducted with a rather optimistic configuration in which the inversion system is perfectly informed about the statistics of the limited number of error sources. These OSSEs indicate that the image resolution has to be finer than 4 km to decrease the uncertainty in the 6-hour mean emissions by more than 50%. More complex experiments assess the impact of more realistic error estimates that current inversion methods do not properly account for, in particular the systematic measurement errors with spatially correlated patterns. These experiments highlight the difficulty to improve current knowledge on CO$_2$ emissions for urban areas like Paris with CO$_2$ observations from satellites, and call for more technological innovations in the remote sensing of vertically integrated columns of CO$_2$ and in the inversion systems that exploit it.





## 1 Introduction

Measurements of $CO_2$ atmospheric concentrations have been used for decades to estimate $CO_2$ natural surface fluxes at global to regional scales based on atmospheric transport modelling and statistical atmospheric inversion techniques (Peylin et al., 2013). There is now a growing political and scientific interest for the atmospheric monitoring of $CO_2$ emissions from

cities (Duren and Miller, 2012). It has encouraged the investigation of space borne imagery techniques for $CO_2$ concentrations at, typically, 2 to 10 km spatial resolution. The idea is that these techniques may have a high potential for characterizing the plumes of $CO_2$ downwind large cities, which could be used to quantify the underlying emissions based on an atmospheric inversion approach. Examples of satellite $CO_2$ imagery concepts are (i) the CarbonSat mission which was a candidate for ESA's Earth Explorer 8 opportunity (Bovensmann et al., 2010; Buchwitz et al., 2013; Sierk et al., 2014), (ii) a

project to measure $CO_2$ with the European Sentinel-5 mission for which a preliminary study of feasibility was conducted but which was not retained (Chimot et al., 2012), (iii) a European satellite dedicated to $CO_2$ anthropogenic emissions that is presently studied by ESA and the European Commission (Ciais et al, 2015) and (iv) the GeoCARB geostationary mission which has been selected as an Earth Venture Mission by NASA (O'Brien et al., 2016). The three first of these concepts rely on $CO_2$ imagery from sun-synchronous and low earth orbit (LEO) satellites based on differential absorption measurements in

the Short Wave Infra Red (SWIR). The SWIR instruments presently appear to be the most suitable for the monitoring of $CO_2$ surface fluxes, because of their sensitivity to $CO_2$ concentrations in the planetary boundary layer, where the signal of these fluxes is the largest (Buchwitz et al., 2015). The proposed swath width of CarbonSat was on the order of 200-300 km with a ~2 km horizontal resolution, whereas that of Sentinel-5 is much larger i.e. ~2000 km with a 7 km horizontal resolution. The swath has a direct impact on the revisit period. These missions cannot sample $CO_2$ in cloudy areas, but, weather permitting,

this revisit period is close to one day for Sentinel-5 and larger than a week for a CarbonSat-like system. These missions should be envisaged as part of constellations (Velazco et al., 2011) and of integrated observation frameworks including in situ networks. However, given the lack of concept regarding the potential synergy between different space missions and in situ networks for monitoring city emissions, there is a need to study the ability of each space mission to bring information on the emissions from individual cities as a stand alone observation system.

There have been attempts to exploit the space borne measurements of vertically integrated columns of dry air mole fractions of $CO_2$ ($XCO_2$) based on SWIR instruments on-board SCIAMACHY or GOSAT for the large scale inverse modelling of $CO_2$ natural fluxes (Buchwitz et al., 2015; Basu et al., 2013; Chevallier et al., 2014). These previous works demonstrate the large detrimental impacts of atmospheric transport model errors, underlying the inversion systems, and of the so-called systematic errors in the $XCO_2$ retrievals. Conversely to the radiometric measurement noise, these systematic errors, which

are connected to uncertainties in the atmospheric radiative transfer, have coherent spatial patterns consistent with variations in, e.g., surface albedo, atmospheric aerosols or thin clouds (Chevallier, 2015). The first releases of $XCO_2$ data from SWIR measurements by OCO-2, which was put into orbit during summer 2014, still bear the signature from such systematic errors



(Eldering et al., 2017). Subsequently, routine estimates of $CO_2$ natural fluxes from atmospheric inversion approaches are still based on the assimilation of the sole ground based in situ measurements (Peylin et al., 2013; Le Quere et al., 2015).

The application of atmospheric inversions for the monitoring of $CO_2$ emissions from cities is a recent research activity. Initial results are based on dedicated urban ground based atmospheric $CO_2$ measurement networks (Breon et al., 2015;

Lauvaux et al., 2013; Turnbull et al., 2015). There has been no attempt at using SCIAMACHY or GOSAT data at this scale due to the low resolution and precision of the SCIAMACHY data or to the scarcity of the sampling by GOSAT. Kort et al. (2012) discussed the possibility to track trends of $CO_2$ emissions from large megacities based on the diagnostics of local enhancements of the GOSAT $XCO_2$ data over these megacities, but they did not perform inversions of their emissions using these data. OCO-2 provides measurements from 8 across track pixels of ~1.3 km by ~1.3 km resolution. Due to such a high

sampling density along the satellite overpass and to their relatively high theoretical precision, OCO-2 data may bring more insight into the $XCO_2$ field nearby the cities than SCIAMACHY and GOSAT, especially when using the specific targeting mode of this mission (Crisp and Team, 2015). However, the relatively narrow swath of OCO-2 hinders a full view of urban emission plumes. Bovensmann et al. (2010), Krings et al. (2011) and Krings et al. (2016) raised optimism regarding the potential of $XCO_2$ imagery for the monitoring of point sources such as power plants or cities. Rayner et al. (2014) and

O'Brien et al. (2016) assessed the potential of the GeoCARB instrument which would provide co-located data of vertically integrated columns of dry air mole fractions of both $CO_2$ and CO (XCO), which is co-emitted with $CO_2$ by the fossil fuel combustion. Their conclusions were less optimistic regarding the potential of the sole imagery of $XCO_2$, indicating that the XCO imagery should provide a better constraint for the estimate of $CO_2$ anthropogenic point sources.

The present study aims at analysing the potential of the $XCO_2$ imagery from a sun-synchronous satellite with a SWIR

radiometer like CarbonSat and Sentinel-5 for the monitoring of $CO_2$ emissions from a megacity. It should also bring some characterization of the potential of the $XCO_2$ imagery from a geostationary mission like GeoCARB. Efforts are put in place to ensure that this analysis is based on a realistic configuration of the emissions and atmospheric transport in and around an existing megacity, and on realistic estimates of the spatial coverage, of the random noise and of the systematic errors from the $XCO_2$ imagery. It is based on a state-of-art atmospheric inversion methodology, which can be called "pixel-based" since

it assimilates the $XCO_2$ data corresponding to each pixel of the satellite image as independent observations.

The Paris urban area has ~11 million people and emits ~11-14 MtC.y$^{-1}$ (Staufer et al., 2016, AIRPARIF 2013). It covers the Paris administrative city which has ~2 million people, and its urban surrounding which has ~9 million people. This urban area is chosen as a study case to benefit from the expertise and tools developed for the assimilation of in situ $CO_2$ measurements from the $CO_2$-Megaparis and ICOS ground based networks (Breon et al., 2015). In particular, the inversions

in the present study are based on a "bottom-up" knowledge of the $CO_2$ emissions at 1 km / 1-hour resolution from the regional inventory established by the AIPARIF air-quality agency (AIRPARIF 2013), and on a 2 km resolution configuration of the CHIMERE atmospheric transport model driven by ECMWF meteorological analysis as in Breon et al. (2015) and Staufer et al. (2016). Furthermore, the Paris urban area is the most populated urban area of Europe and the Paris administrative city is the densest city of Europe. The emissions from this megacity (mainly from transport and heating) are



high and concentrated over a relatively small area. Therefore, it should be a favourable case for the inversion of city emissions based on space-borne imagery. There is no other major city or area of $CO_2$ emissions in its vicinity, which should help distinguishing the plume of Paris from that of other regional sources. Finally, the topography in the city and in its vicinity is relatively flat and the average wind speed in the region at 100 m above the ground level (magl) is ~7 ms$^{-1}$ so that

the plume of $XCO_2$ out of the city should have a relatively simple structure and should be easy to model. By contrast, $CO_2$ plumes or domes of cities in environment with complex terrain, or affected by sea breezes in coastal areas (Perez-Landa et al., 2007) are more difficult to simulate. Consequently, the ability to monitor the emissions from the Paris urban area can be seen as a pre-requirement for the more general task of tracking the emissions from other megacities or smaller cities in Western Europe.

Preliminary simulations and inversion experiments (using the modelling configuration described in section 2.5) demonstrated that, due to atmospheric diffusion, the signature of the emissions at a given time from the Paris urban area is negligible or hardly detectable in the $XCO_2$ fields approximately 6 hours later. Consequently, an $XCO_2$ image from a satellite cannot be exploited to infer direct information about the emissions earlier than ~6 hours before the satellite overpass. Therefore, this study focuses on the analysis of the potential of individual $XCO_2$ images of the Paris area for

inverting the emissions during the 6 hours before each satellite overpass, without trying to exploit successive overpasses together. In order to derive statistics that are representative of a wide range of observation conditions, different experiments are conducted for images taken on different days, i.e., with different meteorological conditions, different cloud cover and different occurrences of the satellite systematic errors.

The study focuses on the ability to derive the average emissions from the city over the 6-hour periods before the satellite

overpasses but also to solve for the temporal variations at hourly resolution during these periods or to solve for the distribution between the main sectors of activity, i.e., traffic, domestic and commercial heating and industrial combustion. This ability to solve for the sectorial distribution is quite equivalent to the ability to solve for some spatial distribution of the emissions since the emissions of each sector of activity have a different spatial distribution.

The potential of the satellite imagery is assessed in terms of improvement of some statistical prior knowledge on the

emissions. Using this statistical prior knowledge, some statistical knowledge on $XCO_2$ derived from the space-borne measurements, and the relationship between the emissions and $XCO_2$ given by an atmospheric transport model for the Paris area, the atmospheric inversion follows the Bayesian theory to derive a new ("posterior") statistical knowledge on the emissions with lower uncertainties than the prior. The improvement of the knowledge on the emissions enabled by the assimilation of the satellite data is thus quantified in terms of uncertainty reduction.

Since the atmospheric measurement networks are generally sparse, traditional atmospheric inversion applications have used prior knowledge on the $CO_2$ fluxes to decrease the uncertainties and gaps in the estimate of the fluxes based on the statistical assimilation of the data from these networks. Prior knowledge is generally based on "bottom-up" models or statistical inventories of the processes underlying the fluxes. At city-scale, and due to the swath of the satellite imagery, one expects that the inversion may derive good estimates of the emissions using this imagery only, or that such estimates could be used





for the independent verification of the "bottom-up" inventories. Even though, strictly speaking, such a purely "top-down" approach is not investigated here, this study can still give insights on the potential for deriving the emissions based on space-borne data only.

The assessment of the uncertainty reduction due to the assimilation of the satellite imagery is based on different Observing
System Simulation Experiment (OSSE) configurations, which account for the time, location, spatial resolution and error statistics of the $XCO_2$ measurements associated with the satellite and instrumental configurations.

Section 2 details the theoretical and practical inversion framework and the principles for the assessment of the potential of the satellite imagery as a function of its design. Section 3 analyses an 'optimistic' assessment of this potential based on a configuration ignoring sources of errors hardly accounted for by state-of-the-art inversion systems, such as biases in the
observation operator or observation errors whose spatial correlations have complex patterns. Section 4 diagnoses the impact of such errors. The discussions and conclusions in section 5 confront the optimistic and pessimistic assessments in order to infer the robustness and extent of the conclusions that can be derived with state-of-the-art systems regarding the potential of $XCO_2$ imagery.

## 2 OSSE framework

The flux uncertainty reduction due to the assimilation of satellite images is defined as the relative difference between the prior and posterior uncertainties in the fluxes. Its statistical characterization is double: the statistics of the posterior uncertainties are provided for a given day following the statistical nature of the inversion framework, but they are also sampled for different days i.e. different atmospheric transport conditions.

For a given day, the atmospheric inversion derives a statistical estimate for a set of n input parameters $\mathbf{s} = [s_1 \dots s_n]^T$ of a
linear coupled flux and atmospheric transport model $\mathbf{s} \rightarrow \mathbf{y} = \mathbf{M s} + \mathbf{y}^{\text{fixed}}$ that simulates the satellite image of $XCO_2$ $\mathbf{y}$. The coupled model is called observation operator hereafter. This derivation is based on the information brought by the measurements $\mathbf{y}^o$. The control parameters underlying the Paris emissions are the target quantities. Other control parameters correspond to sources of uncertainties in the observation operator that can be better handled if explicitly solved for, e.g., the ecosystem fluxes in the region. The other sources of uncertainties and errors affecting the misfits between the model
simulations and the measurements that are not controlled by the inversion should be accounted for as errors in the observation space that the inversion attempts at filtering out. In OSSEs, ignoring some of the sources of uncertainties both in the control space and in the observation space yields optimistic results. By controlling the hourly temporal variations or the sectorial distribution of the emissions, the system accounts for the impact of uncertainties in these distributions on the retrieval of 6-hour mean emissions before a satellite overpass. Table 1 summarizes the different options taken for the
configuration of the OSSEs and Table 2 summarizes the different experiments based on various combinations of these options.



### 2.1 Spatial domain

The study is based on a regional atmospheric transport configuration with input $CO_2$ conditions at the boundaries of its domain. The simulation domain is illustrated in figure 1. It is approximately centred over the Paris urban area and it encompasses a large part of Northern France. This domain is the same as that used in Breon et al. (2015). It is sufficiently

large (~500 km × 500 km) so that the $CO_2$ emitted within Paris at a given time does not exit the simulation domain within 6 hours, at least with the wind conditions considered in this study. Therefore, in this study, the signature of the emissions from Paris in the $XCO_2$ field becomes undetectable in the satellite images as a result of the atmospheric transport diffusion or by leaving the swath of a satellite. It does not become undetectable artificially by leaving the simulation domain, which would have biased the evaluation of large swath satellite configurations.

### 2.2 Temporal framework

The satellite observation is assumed to occur at 11:00 local time in the morning, in line with the CarbonSat mission requirements (ESA, 2015). In the Paris area, local and UTC time, which is used hereafter, have a ~9 min 30 s difference, which is negligible. Therefore, the study focuses on the ability to retrieve information on the emissions between 5:00 and 11:00 based on a satellite image acquired at 11:00.

The results are investigated for 20 different days of October 2010, i.e., 20 different transport conditions and 20 different $CO_2$ domain boundary conditions. These 20 different days encompass a wide range of wind conditions that strongly influence the amplitude, shape and orientation of the $XCO_2$ plume. They are called hereafter inversion days. In October, the ecosystem fluxes are rather weak which, in principle, should facilitate the separation between the atmospheric signatures of the natural fluxes and anthropogenic emissions and thus the estimate of the anthropogenic emissions from the $XCO_2$ imagery.

### 2.3 Observation space **y**

In the following, we consider instruments with a very good point spread function, so that their measurements are representative of their horizontal sampling, and so that this horizontal sampling is fully characterized by the spatial resolution. The expression "centred on Paris" is used to indicate that an area is centred on the centre of Paris.

ESA (2015) presents two concepts for CarbonSat with 240 km and 185 km swaths, and identifies 240 km as a breakthrough

requirement for the swath of the mission. Since it accounts for the design of the CarbonSat and Sentinel-5 missions, this study investigates three types of $XCO_2$ spatial sampling in the Paris area (see Table 1).

The "TH-CarbonSat" sampling is a full sampling of a 150 km radius circle centred on Paris at 2 km resolution that ignores cloud cover. This theoretical sampling corresponds to an optimistic configuration of the CarbonSat mission with a 300 km swath and to an optimal satellite trajectory for the monitoring of the emissions of Paris. This sampling will be used for

optimistic experiments only.





The "TH-LargeSwath" sampling is a full sampling of the whole simulation domain at 4 km, 6 km, 8 km or 10 km resolution, which ignores cloud cover. This theoretical sampling simulates the acquisition by an instrument with a larger swath but a smaller spatial resolution than CarbonSat, such as the 7 km resolution instrument on-board Sentinel 5. This sampling will also be used for optimistic experiments only.

The "SIM-CarbonSat" samplings (see figures S1 and S2) correspond to realistic simulations of the sampling of the Paris area by a 240 km swath instrument with 2 km resolution such as CarbonSat. accounting for cloud cover. They are extracted from simulations of CarbonSat sampling over the whole globe and for a full year by Buchwitz et al. (2013). Pillai et al. (2016) also used these simulations to model CarbonSat observations with 240 km and 500 km swaths. From these simulations, 69 CarbonSat passes in our modelling domain over the year provide at least 1 $XCO_2$ data in the 100 km radius circle centred on

Paris and are considered to be "over the Paris area". 19 different cases are selected from these 69 passes for each of the 20 inversion days in October 2010. These are the 19 simulated passes over the Paris area that provide the best constraints on the emission inversion. This ranking follows the theoretical uncertainty reductions associated with the assimilation of the different sets of observations from the passes as detailed in sections 2.8.3 and 4.2. It depends on the number of 2 km × 2 km observations and on their position relative to the emission atmospheric plume, i.e., on the cloud cover and the swath position

with respect to the Paris target associated with a pass over the Paris area. Section 4.2 will show that for any inversion day in October, and thus for any of the modelled emission plumes, the constraint on the emission inversion of the 20[th] best-simulated pass is rather weak. This explains why only 19 simulated passes are selected. The "SIM-CarbonSat" samplings will be used in the less optimistic experiments only. By construction, the CarbonSat acquisitions that are tested for a given inversion day in October 2010 were simulated for other days of the year 2010 by Buchwitz et al. (2013). This is not

identified as a significant issue, even though the atmospheric transport conditions of the inversion days could have some inconsistencies with the cloud cover underlying the simulation of these CarbonSat acquisitions.

**2.4 Control vector s**

Hereafter, the $CO_2$ emissions from anthropogenic activities in the Paris area are denoted: Fossil Fuel (FF) fluxes whereas the natural fluxes in the simulation domain are identified as Net Ecosystem Exchange (NEE). The OSSEs investigate the ability

of the satellite imagery to solve either for the hourly variations or for the sectorial distribution of the 6-hour emissions, and the uncertainties in the estimates of 6-hour mean emissions due to uncertainties in either these temporal or sectorial distributions. Two configurations of the control vector are thus used for a given day: either a control vector solving for the hourly fluxes and the background concentration (see its definition below) with 13 parameters (6 for both hourly FF fluxes and hourly NEE and 1 for the background concentration):

$$\mathbf{s} = [\lambda_5^{FF}, \lambda_6^{FF}, \dots, \lambda_{10}^{FF}, \lambda_5^{NEE}, \lambda_6^{NEE}, \dots, \lambda_{10}^{NEE}, b]^T \qquad (1)$$

or a control vector solving for the 6-hour budgets of the FF $CO_2$ emissions from transportation, residential combustion, commercial combustion and industrial combustion, and for the distribution of the 6-hour budgets of NEE per main Plant





Functional Types (PFT: broadleaf forest, needle leaf forests, grasslands, croplands) with 9 parameters (4 for both the sectorial budgets of FF fluxes and the budgets of NEE per land cover type, and 1 for the background concentration):

$$\mathbf{s} = \left[\lambda^{FF}_{Transportation}, \lambda^{FF}_{Res.\ Combust.}, \lambda^{FF}_{Comm.\ Combust.}, \lambda^{FF}_{Large\ Indust.}, \lambda^{NEE}_{Cropland}, \lambda^{NEE}_{Grassland}, \lambda^{NEE}_{NeedleLeaf\ F.}, \lambda^{NEE}_{BroadLeaf\ F.}, b\right]^{T} \quad (2)$$

In these formal definition of the control vector, $\lambda^{X}_{hh}$ (first case) or $\lambda^{X}_{Y}$ (second case) is a scaling factor for the total flux X

(NEE or FF) between time $hh$ and $hh+1$ hour (first case) or for sector / PFT Y between 5:00 and 11:00 (second case). Almost all of the Paris emissions are distributed among the four FF emission sectors considered in the second configuration of the control vector (see section 2.5.1). With both configurations, controlling scaling factors is equivalent to controlling the budget of the corresponding flux components. In practice, these control parameters are used to rescale the 1-hour and 2 km resolution FF and NEE fields corresponding to these components from the Airparif inventory and an ecosystem model (see

below the description of the observation operator in section 2.5). The emission maps and the hourly budgets from the Airparif inventory vary in time at the hourly scale (see section 2.5.1). Therefore, if the inversion derives identical scaling factors for different hours, it would not lead to identical estimates of the emission budgets for these different hours. However, the variations of the hourly budgets from the Airparif inventory are sufficiently small to be ignored when comparing uncertainties in scaling factors for different hours in the following. $b$ is the average $XCO_2$ concentration in the

simulation domain at the time of the satellite observation (11:00) due to the very large scale influence of $CO_2$ fluxes outside the domain or before 5:00. Its inclusion in the control vector provides a simple account for uncertainties in these remote fluxes. It is thus called "background $XCO_2$" throughout the text. The general notation $\mathbf{s} = [\boldsymbol{\lambda}, b]^{T}$ is used for both control vectors.

The analysis mainly focuses on results obtained for the control variables related to the Paris emissions which are the target quantities. The control of the NEE could also bring some insights about the inversion of the natural fluxes using satellite data but this topic is out of the scope of this paper and the experimental framework is not optimized for such analysis.

### 2.5 The observation operator

The linear observation operator $\mathbf{s} \rightarrow \mathbf{y} = \mathbf{M}\,\mathbf{s} + \mathbf{y}^{\text{fixed}}$ combines three linear operators.

### 2.5.1 Spatial and temporal distribution of the fluxes

The first operator is the rescaling of 2 km and 1-hour resolution fields of emissions from Paris and Northern France NEE to generate the fields of total fluxes $\mathbf{f}$ in the domain: $\boldsymbol{\lambda} \rightarrow \mathbf{f} = \mathbf{M}_{\text{inventory}}\,\boldsymbol{\lambda}$. The emission and NEE fields are based on estimates from the Airparif inventory for the year 2008 and from simulations with the C-TESSEL vegetation carbon flux model at ECMWF (Boussetta et al., 2013). In order to fully describe the total fluxes in the domain, one must also account for the anthropogenic emissions outside the Paris area that are not covered by the Airparif inventory. However, the OSSEs



ignore uncertainties in these emissions (see the corresponding discussion in section 2.9). Consequently, there is no term associated with these emissions in the equations used in this study and they are ignored in the analysis of the results.

Breon et al. (2015) also used the Airparif inventory for the year 2008 and the C-TESSEL NEE simulations in their inversion framework. They provided some description and analysis of these products, which are not reminded here. For each inversion

day in October 2010, we use the estimates from Airparif for the corresponding day in October 2008 and that from C-TESSEL for the same day in October 2010. The inconsistency between the year of emission estimates and that of the transport simulation is not an issue for the OSSEs.

The resolution of the Airparif inventory is 1 km in space and 1-hour in time, and the C-TESSEL NEE is provided at approximately 15 km and 3-hour resolution. These products are aggregated or interpolated at 2 km and 1-hour resolution,

which is the input resolution of the transport model. The sectorial resolution of the Airparif inventory is much more detailed than that considered with the second configuration of the control vector, which rescales only four main aggregated sectors, representing together more than 95% of the total emissions. When using this second type of control vector, the NEE from C-TESSEL is interpolated and then disaggregated in each 2 km × 2 km grid cell between the PFT components of the control vector. Since, we do not have access to estimates of the NEE per PFT by C-TESSEL, this disaggregation is based on the

fractional coverage of each PFT per grid cell, derived from the land cover distribution of the Global Land Cover Facility (GLCF) 1 × 1 km resolution database from the University of Maryland (Hansen and Reed, 2000).

### 2.5.2 Atmospheric transport

The second operator is the atmospheric transport of $CO_2$ in the study domain that yields the temporal and 3D spatial distribution of $CO_2$ atmospheric mole fractions $\mathbf{c}$: $\mathbf{f} \rightarrow \mathbf{c} = \mathbf{M}_{transport}\, \mathbf{f} + \mathbf{c}^{bound}$. This operator is based on the CHIMERE

atmospheric transport model (Menut et al., 2013). The term $\mathbf{c}^{bound}$ contains the signature of the domain boundary and initial (at 5:00) $CO_2$ conditions, called hereafter "boundary conditions". $\mathbf{c}^{bound}$ is decomposed into $b\,[1,1 \ldots 1]_{n_c}^{T}$ and $\mathbf{c}^{fixed}$, where $[1,1 \ldots 1]_{n_c}$ has the length $n_c$ corresponding to the 1D representation of the 3D CO2 concentrations space, $\mathbf{c}^{fixed}$ is not controlled by the inversion, and the vertical (see section 2.5.3) and horizontal integration of $\mathbf{c}^{fixed}$ within the simulation domain at 11:00 is null, so that the vertical and horizontal integration of $\mathbf{c}^{bound}$ at 11:00 yields $b$.

The CHIMERE configuration used in this study is similar to that used in Breon et al. (2015), e.g. it has the same vertical discretization with 19 vertical levels from the surface up to 500 hPa ($P_{top}$) and the same physical parameters. It has a 2 km horizontal resolution over the whole simulation domain. ECMWF operational analyses at nearly 15 km resolution are used to provide the meteorological forcing to CHIMERE. This meteorological forcing does not account for urban land surface influences but we may neglect them for the OSSEs considered here.

The $CO_2$ mixing ratios at the lateral and top boundaries of the CHIMERE model and the initial conditions at 5:00 (i.e. $\mathbf{c}^{fixed}$) are imposed using outputs from a global LMDZ simulation at 3.75° (longitude) × 2.5° (latitude) resolution assimilating in situ $CO_2$ data from a global ground based measurement network (Chevallier et al., 2010).



### 2.5.3 Vertical integration of the CO₂ column

The third operator is the computation of XCO₂ data **y** for a given satellite observation sampling based on the 3D CO₂ concentrations at 11:00 in the CHIMERE domain, from the surface up to $P_{top}$: $\mathbf{c} \rightarrow \mathbf{y} = \mathbf{M}_{integ}\, \mathbf{c}$. Part of this computation includes the aggregation of the 2 km horizontal resolution CO₂ fields at the chosen satellite image resolution. For the sake of

simplicity and since we use synthetic data only, it is assumed that the satellite observation has a uniform vertical weighting function for each horizontal pixel of a given satellite space sampling (see above section 2.3). For a given horizontal pixel at latitude *lat* and longitude *lon*, XCO₂ is thus computed from the average of the vertically distributed CO₂ mole fractions:

$$XCO_2(lon,lat) = \frac{1}{P_{surf}(lon,lat)}\left(\int_{P_{top}}^{P_{surf}(lon,lat)} CO_2(lon,lat,p)dp + \overline{CO_2}\left(P_{top}\right)P_{top}\right) \qquad (3)$$

where $p$ is the pressure and $P_{surf}$ is the surface pressure. In this equation, a uniform concentration equal to the horizontal

average of the top level mixing ratios in CHIMERE: $\overline{CO_2}\left(P_{top}\right)$ is assumed to apply at all pressures lower than $P_{top}$. This simple approximation is used since the surface fluxes in the simulation domain do not impact the concentration close to the top of the model within the simulation time. It ignores the signature of remote fluxes above 500 hPa and we implicitly assume that this signature will not significantly impact the inversion of the emissions from the Paris area. This equation also ignores the slant path of the columns that would be measured by a satellite within the model. Finally it ignores the potential

impact of the water vapour content of the atmosphere on the vertical weighting function of the satellite measurements.

### 2.5.4 Practical computation of the observation operator

The observation operator $\mathbf{s} \rightarrow \mathbf{y} = \mathbf{M}\,\mathbf{s} + \mathbf{y}^{fixed}$ can be rewritten $\mathbf{s} \rightarrow \mathbf{y} = \mathbf{M}_{integ}\, \mathbf{M}_{transport}\, \mathbf{M}_{inventory}\, \boldsymbol{\lambda} + b\,[1,1\,...\,1]_{n_y}^{\,T} + \mathbf{y}^{fixed}$ where $[1,1\,...\,1]_{n_y}$ has the length $n_y$ corresponding to the 1D representation of the observation space. In order to apply the inversion equations analytically (see section 2.6 below), the

**M** matrix is built explicitly. For this, the $\mathbf{M}_{integ}\, \mathbf{M}_{transport}\, \mathbf{M}_{inventory}$ matrix is built by computing each of its column $\mathbf{m}_j$ through the application of the operator $\boldsymbol{\lambda} \rightarrow \mathbf{y} = \mathbf{M}_{integ}\, \mathbf{M}_{transport}\, \mathbf{M}_{inventory}\, \boldsymbol{\lambda}$ to unit vectors $\boldsymbol{\lambda}_j = [0\,...\,0\,1\,0\,...\,0]^T$ where only the j$^{th}$ control parameter corresponding to a flux scaling factor is set to 1 and others are set to 0. The small number of control parameters makes the number of simulations $\boldsymbol{\lambda}_j \rightarrow \mathbf{m}_j = \mathbf{M}_{integ}\, \mathbf{M}_{transport}\, \mathbf{M}_{inventory}\, \boldsymbol{\lambda}_j$ computationally affordable. The columns $\mathbf{m}_j$ correspond to the response function in terms of XCO₂ to each of the Airparif or C-TESSEL flux component

controlled by a scaling factor. The full observation operator matrix **M** is obtained by adding the column $[1,1\,...\,1]_{n_y}^{\,T}$ to $\mathbf{M}_{integ}\, \mathbf{M}_{transport}\, \mathbf{M}_{inventory}$ which corresponds to the homogeneous distribution in space of $b$, the background XCO₂ at 11:00.



### 2.6 Theoretical framework of the atmospheric inversion

The theoretical framework of the inversion system used here in the OSSEs is the one traditionally used for atmospheric inversions. It is based on the Bayesian update of a statistical prior estimate of the control vector $\mathbf{s}$, using statistical information from the assimilation of the measurements $\mathbf{y}^o$ in the observation operator. The usual assumption is that the prior

estimate has a Gaussian distribution $N(\mathbf{s}^b, \mathbf{B})$ and that the distribution of the misfits between the simulated observations $\mathbf{M}\,\mathbf{s} + \mathbf{y}^{fixed}$ and $\mathbf{y}^o$ that are not due to errors in $\mathbf{s}$, i.e. the so-called observation errors, which include atmospheric transport and representation errors, is unbiased, has a Gaussian distribution $N(\mathbf{0}, \mathbf{R})$, and is not correlated with the prior uncertainty. In that case, the Bayesian update of the estimate of $\mathbf{s}$, called hereafter the posterior distribution, is a Gaussian distribution $N(\mathbf{s}^a, \mathbf{A})$ with $\mathbf{s}^a$ being the best estimate of the actual $\mathbf{s}$ knowing $\mathbf{s}^b$ and $\mathbf{y}^o$, and $\mathbf{A}$ characterizing the uncertainty in this

estimate. The problem simplifies into deriving (Tarantola, 2005):

$$\mathbf{A} = [\mathbf{B}^{-1} + \mathbf{M}^T \mathbf{R}^{-1} \mathbf{M}]^{-1} \tag{4}$$

and

$$\mathbf{s}^a = \mathbf{s}^b + \mathbf{K}\left(\mathbf{y}^o - \mathbf{M}\,\mathbf{s}^b - \mathbf{y}^{fixed}\right) \tag{5}$$

by denoting

$$\mathbf{K} = \mathbf{A}\,\mathbf{M}^T\,\mathbf{R}^{-1} \tag{6}$$

The inversion system solves for these equations analytically based on the explicit computation of all matrices and vectors.

Hereafter, we characterize the uncertainties by their standard deviations and correlations. The scores of "uncertainty" and "uncertainty reductions" that we give for a given scalar quantity, a single flux budget or parameter, refer to the standard deviation of the uncertainties $\boldsymbol{\sigma}$ in this scalar quantity, and to the relative difference between its prior and posterior values: 1-$\sigma_a/\sigma_b$. An exception occurs when explicitly accounting for biases (see sections 2.8.2 and 2.8.3 below), in which case the scores of "uncertainty" and "uncertainty reductions" refer to the Root Mean Square (RMS) values of the uncertainties and to

the relative difference between these prior and posterior RMS values.

### 2.7 Prior uncertainty

In all cases, our framework assumes that the prior uncertainty in individual scaling factors for the fluxes i.e. the relative uncertainty in the budget of the corresponding flux components from the observation operator is 50%, that there is no correlation between uncertainties in the different scaling factors and that the uncertainty in $b$ is 10 ppm. $\mathbf{B}$ can thus be

written $\mathbf{B} = \mathrm{diag}([0.5^2, 0.5^2...0.5^2, 10^2\ \mathrm{ppm}^2]^T)$ denoting hereafter diag($\mathbf{v}$) for a diagonal matrix whose diagonal is defined by the terms in vector $\mathbf{v}$. This results in a small inconsistency regarding the uncertainty in the total 6-hour mean emissions from Paris when controlling the hourly fluxes or the fluxes for the different sectors of activity: it is nearly equal to 22.4% or 26% respectively, even though it also slightly depends on the days. Such a difference depending on the control vector also applies to the 6-hour mean NEE. These differences are negligible in the framework of the OSSEs considered here. The

sensitivity of the results to the control vector configuration is mainly indicative of the impact of uncertainties in the temporal



profile vs. the sectorial distribution of the emissions, and not of the impact of changes of the prior uncertainty in the 6-hour mean emissions.

### 2.8 Observation errors and practical implementation of the OSSEs

Tables 1 and 2 summarize the different configurations of the OSSEs. Different types of OSSEs are conducted with different

levels of agreement between the statistics of the synthetic observation errors that are introduced in practice and the assumptions underlying the theoretical framework of the inversion system.

#### 2.8.1 Analytical computation of the posterior uncertainties when considering only a measurement noise that is perfectly accounted for by the inversion system

In the first set of OSSEs, errors from the observation operator are ignored and the observation errors are limited to a

Gaussian noise in the measurements. The standard deviation of this noise is perfectly consistent with the configuration of the $\mathbf{R}$ matrix in the inversion system. In this case, the assumptions made through the set-up of the inversion system are exact, and the matrix $\mathbf{A}$ obtained from the application of equation 4 is a perfect estimate of the uncertainties in the inverted fluxes. $\mathbf{A}$ and its comparison to $\mathbf{B}$, i.e., the analysis of the so-called uncertainty reduction due to the assimilation of the measurements, are the proper indicators of the satellite data potential. Since $\mathbf{A}$ depends on $\mathbf{B}$, $\mathbf{R}$ and $\mathbf{M}$ only, the building of

synthetic data is unnecessary. The corresponding OSSEs will use the two first types of satellite spatial sampling described in section 2.3 only (TH-CarbonSat and TH-LargeSwath). A uniform value that does not vary in space or time for a given satellite configuration is used for the standard deviation of the measurement noise. This ignores that, in principle, the standard deviation of the measurement noise should vary significantly with the solar zenith angle, the surface albedo and the atmospheric content. For TH-CarbonSat sampling, this uniform value is derived as a typical value from the simulations of

CarbonSat random errors over the Paris area performed by Buchwitz et al. (2013). For the TH-LargeSwath sampling, the values are derived from the ESA Sentinel 5 study (Chimot et al. 2012) with two options corresponding to hypothetical instruments with one or two SWIR bands: the 2.0 µm and optionally the 1.6 µm absorption bands. $\mathbf{R}$ is thus set-up as $(e^r)^2 \mathbf{I_d}$ where $e^r$ ranges from 1.1 ppm to 2.1 ppm depending on the satellite configuration (see Table 1) and $\mathbf{I_d}$ is the identity matrix in the observation space.

All inversions solving for the fluxes per sector of activity and per PFT will be conducted with this OSSE framework and with these assumptions regarding the observation errors. The other types of OSSEs and assumptions on the observations errors will be tested with the inversion of hourly emissions only. This choice is based on the analysis of the first set of OSSEs commented in sections 3.2 and 3.3.

#### 2.8.2 Analytical computation of the impact of biases from the observation operator

In the second set of OSSEs, biases $\mathbf{y}^{\text{bias}}$ from the observation operator and thus in the observation errors are introduced in addition to the measurement noise. They impact the difference $\mathbf{y}^0 - \mathbf{M}\,\mathbf{s}^b - \mathbf{y}^{\text{fixed}}$. Such biases cannot be accounted for by





the theoretical framework of the inversion systems and generate a bias in the posterior estimate of the control vector equal to:

$$\mathbf{s}^{\mathrm{bias}} = \mathbf{K}\,\mathbf{y}^{\mathrm{bias}} \tag{7}$$

This bias will be computed along with A and both this bias and A will have to be compared to B in order to infer the potential

of the satellite data assimilation. The second set of OSSEs will use the two first types of satellite spatial sampling only (TH-CarbonSat and TH-LargeSwath) and the value for R=$(e^r)^2 I_d$ as defined above for the first set of OSSEs.

Two types of biases in the observation operator will be investigated. The first type of bias is related to the spatial distribution of the emissions of Paris. The operator $\mathbf{M}_{\mathrm{inventory}}$ used by the inversion system could strongly differ from the actual distribution of the emissions $\mathbf{M}_{\mathrm{inventory}}^{\mathrm{true}}$. Assuming that this difference is a bias and not a random error, which makes sense

given the relatively weak temporal variability of the spatial distribution of the emissions, one can easily demonstrate that this generates a bias in the estimate of the true control vector $\mathbf{s}^{\mathrm{true}}$, which is given by

$$\mathbf{s}^{\mathrm{bias}} = \mathbf{K}\left(\mathbf{M}_{\mathrm{inventory}}^{\mathrm{true}} - \mathbf{M}_{\mathrm{inventory}}\right)\mathbf{s}^{\mathrm{true}} \tag{8}$$

This does not yield any additional random posterior uncertainty compared to that characterized by the A covariance matrix derived from equation 4. We thus associate it with a bias in the observation errors:

$$\mathbf{y}^{\mathrm{bias}} = \left(\mathbf{M}_{\mathrm{inventory}}^{\mathrm{true}} - \mathbf{M}_{\mathrm{inventory}}\right)\mathbf{s}^{\mathrm{true}} \tag{9}$$

In an extreme case where the spatial distribution of the emissions in the Paris area is fully ignored, one could distribute the emissions homogeneously over a disk whose radius would be approximately that of the Paris urban cover. The difference between the homogeneous distribution over a disk and the distribution from Airparif is used here to simulate a bias in an observation operator that would be built on rough information about the urban extent only.

For the OSSEs investigating the impact of such a bias in the emission spatial distribution, two new observation operators have been computed by replacing the spatial distribution of the Paris emissions from Airparif by a homogeneous one over two different disks centred on Paris with 15 km and 45 km radius respectively (denoting $\mathbf{M}_{\mathrm{inventory}} = \mathbf{M}_{15}^{\mathrm{disc}}$ and $\mathbf{M}_{45}^{\mathrm{disc}}$ respectively), but keeping the same hourly emission budgets as in Airparif. These two radiuses are defined so that the first disk is smaller than the Paris urban area while the second one encompasses all this area. The bias is computed with $\mathbf{s}^{\mathrm{true}} =$

$[\boldsymbol{\lambda}^{\mathrm{true}}, b^{LMDZ}]^T$ where $b^{LMDZ}$ is set according to the average of the LMDZ simulation used to generate $\mathbf{c}^{\mathrm{fixed}}$ (see section 2.5.2) and where $\boldsymbol{\lambda}^{\mathrm{true}} = [1,1\dots1]_{n_\lambda}^{T}$, i.e., assuming that the LMDZ provides the true background $XCO_2$ and that Airparif has the true estimates of the hourly total emissions. This is relevant since this second type of OSSEs aims at deriving a qualitative assessment rather than a precise quantification of the impact of biases in the observation operator.

Of note is the fact that this modification of the observation operator impacts, in practice, equation 4 and thus the theoretical

posterior uncertainties. The $\mathbf{A}$ matrices obtained with $\mathbf{M}_{15}^{\mathrm{disc}}$ and $\mathbf{M}_{45}^{\mathrm{disc}}$ differ from that obtained with $\mathbf{M}_{\mathrm{inventory}}$ in the first



set of OSSEs which relied on a perfect knowledge of the spatial distribution of the emissions. It is thus critical to analyse this change of the random posterior uncertainty.

The second type of bias in the observation operator investigated in this study is connected to errors in the estimate of the $CO_2$ boundary conditions, i.e., in the term $\mathbf{y}^{\text{fixed}} = \mathbf{M}_{\text{integ}} \, \mathbf{c}^{\text{fixed}}$ defined in section 2.5. The uncertainty in the boundary

conditions would be traditionally considered as random in the usual mathematical framework of the inversions. Here, the average impact of the boundary conditions is implicitly assumed to bear random errors through the control of the background concentrations $b$ and the definition of their prior uncertainties. However, the boundary conditions from a very large-scale simulation that misses the right structure of $CO_2$ transported from large and heterogeneous sources and sinks close to the boundaries, e.g. the anthropogenic emissions in Belgium and in the Netherlands, can bear biases. Tests are

conducted by assuming that the bias in $\mathbf{c}^{\text{fixed}}$ is given by the transport with CHIMERE of the variations in space and time of the boundary conditions from the LMDZ simulation around their mean during the 6-hour simulation for a given day in October. $\mathbf{y}^{\text{bias}}$ is given by the projection of this bias in the observation space through $\mathbf{M}_{\text{integ}}$. Removing the mean ensures reasonable amplitude for the bias. It also ensures that we do not account twice for errors in the homogeneous background for $XCO_2$, given that the prior uncertainties in $b$ are already included in the inversion framework.

**2.8.3 Monte Carlo sampling of the posterior uncertainties when introducing systematic measurement errors that have a non Gaussian distribution**

In the third set of OSSEs, the systematic measurement errors are introduced in addition to the random measurement noise but model errors from the observation operator are ignored. The spatial variations of the standard deviation of the measurement noise are also accounted for. The distribution of the systematic errors and thus that of the resulting observation

errors $\{\mathbf{e}_y\}$ is not considered to be Gaussian and it is biased. Such a distribution cannot be perfectly accounted for by the inversion system. The configuration of $\mathbf{R}$ in the inversion system is adapted to fit as much as possible with the distribution of the observation errors given as input to the system. However, in this context, the posterior distribution of errors to the actual control parameters $\{\mathbf{e}_s^a\}$ does not follow the $N(\mathbf{0},\mathbf{A})$ distribution. A Monte Carlo framework is thus set-up to sample $\{\mathbf{e}_s^a\}$ based on the sampling of $\{\mathbf{e}_y\}$ and on an ensemble of applications, for each member of the sampling, of:

$$\mathbf{e}_s^a = \mathbf{e}_s^b + \mathbf{K}\big(\mathbf{e}_y - \mathbf{M}\,\mathbf{e}_s^b\big) \tag{10}$$

This equation 10 is derived by removing the true control parameters in both sides of equation 5, and by denoting the prior distribution of errors to the actual control parameters $\{\mathbf{e}_s^b\}$.

The corresponding Monte Carlo experiments use only the most realistic spatial sampling of the satellite imagery corresponding to the CarbonSat configuration (SIM-CarbonSat) since they rely on the simulation of random and systematic

errors for CarbonSat by Buchwitz et al. (2013). These simulations have been also been used by Pillai et al. (2016) to assess the potential of CarbonSat for monitoring city-scale $CO_2$ emissions. For a given inversion day, the different simulations of



these errors for the different selected passes over the Paris area are used to generate the sampling of measurement errors. This follows the assumption that the statistics of these errors simulated for different days of the year 2010 all follow a single stationary distribution of the measurement error that can apply to any inversion day in October 2010. The OSSEs will thus also have to account for the statistical nature of the cloud cover and of the satellite position. Consequently, each member of

the Monte Carlo ensemble has its own structure for $\mathbf{y}$ and consistently for $\mathbf{M}$, $\mathbf{R}$ and $\mathbf{K}$ so that the application of equation 10 is actually rewritten for the $i^{th}$ member of the ensemble:

$$\mathbf{e}_s^a[i] = \mathbf{e}_s^b[i] + \mathbf{K}[i]\big(\mathbf{e}_y[i] - \mathbf{M}[i]\,\mathbf{e}_s^b[i]\big) \qquad (11)$$

for $i \in [1..N]$ where $\mathbf{K}[i] = \mathbf{A}[i]\,\mathbf{M}[i]^T\,\mathbf{R}[i]^{-1}$ and $\mathbf{A}[i] = \big[\mathbf{B}^{-1} + \mathbf{M}[i]^T\mathbf{R}[i]^{-1}\mathbf{M}[i]\big]^{-1}$

Errors $\mathbf{e}_s^b[i]$ are sampled from the normal distribution N($\mathbf{0}$,$\mathbf{B}$). The statistics of the actual mean $\mathbf{s}_{pract}^{bias}$ and covariance $\mathbf{A}_{pract}$

of the posterior errors $\mathbf{e}_s^a[i]$ are computed to characterize the actual uncertainty reduction compared to $\mathbf{B}$. $\mathbf{s}_{pract}^{bias}$ and $\mathbf{A}_{pract}$ are called "practical" hereafter, in opposition with the theoretical estimates $\mathbf{s}^{bias}$ and $\mathbf{A}$ from equations 4 and 7.

The derivation of the observation errors $\mathbf{e}_y[i]$ and of the corresponding $\mathbf{R}[i]$ matrix is based on the fact that the simulations of random and systematic errors provided by Buchwitz et al. (2013) can be interpreted as maps of the standard deviation $\boldsymbol{\sigma}_y^r[i]$ of the random error $\mathbf{e}_y^r[i]$ and values for the systematic errors $\mathbf{e}_y^s[i]$. These maps also characterize the structure of $\mathbf{y}[i]$. The

maps of systematic errors for the 19 passes of CarbonSat over Paris with the highest number of cloud free pixels at less than 100 km from the centre of Paris are given in figure S1 of the supplementary material. An example of the perturbations of the XCO$_2$ images with such maps of random and systematic errors is given in figure S2. The error $\mathbf{e}_y[i]$ for a given ensemble member is thus built using one of such couple of maps for $\boldsymbol{\sigma}_y^r[i]$ and $\mathbf{e}_y^s[i]$, as the sum of $\mathbf{e}_y^s[i]$ and of a random sample $\mathbf{e}_y^r[i]$ of the normal distribution N($\mathbf{0}$, diag($\boldsymbol{\sigma}_y^r[i]^2$)). As generally done for atmospheric inversion, the matrix $\mathbf{R}[i]$ is deliberately

built as a diagonal matrix, and thus based on the assumption that there is no temporal or spatial correlation in the observation errors, to ensure that it is easily inverted in equation 4. It is derived as the sum of diag($\boldsymbol{\sigma}_y^r[i]^2$) + ($e^s$)$^2\mathbf{I_d}[i]$ where $\mathbf{I_d}[i]$ is the identity matrix for the $\mathbf{y}[i]$ space and $e^s$ is fixed to a typical value for systematic errors equal to 0.3 ppm. This 0.3 ppm value is based on a space and time RMS of the systematic errors over the Paris area in 2010. The addition of the term ($e^s$)$^2\mathbf{I_d}[i]$ helps the inversion system to anticipate for the systematic errors. However, it cannot perfectly anticipate the structure of

these errors due to its inability to account for non Gaussian and biased errors, and for spatial correlations of the errors. In particular, the complex spatial patterns in the systematic errors, e.g., their correlation to aerosol layers or to the variations in land cover, are a critical issue for the inversion.

In this experimental protocol, the size of the Monte Carlo ensembles is limited by the number of selected maps of measurement errors from Buchwitz et al. (2013) i.e. to N=19 members for each inversion day. The theoretical uncertainty

reduction for the 6-hour mean emission estimates for each ensemble member given by the comparison between $\mathbf{A}[i]$ and $\mathbf{B}$ is the criteria for the selection of the "19 best observation sampling" used in this ensemble for a given inversion day. The other





observation samplings generally have very few data on the city emission plume (see section 4.2) while we apply the Monte Carlo framework to derive typical scores of uncertainty reduction for cases when the observation sampling can be assumed to be sufficient to support the emissions inversion. This is why we do not increase the size of the Monte Carlo ensemble by including some of these low observation samplings. However, this modest number of ensemble members can raise sampling

errors (called hereafter Monte Carlo sampling errors) and the ensembles $\mathbf{e}_s^b[i]$ and $\mathbf{e}_s^a[i]$ may not fully converge towards $N(\mathbf{0},\mathbf{B})$ and towards the practical statistics of the posterior uncertainty that are looked for.

There is a need to evaluate the impact of these sampling errors and of the realistic simulation of the observation samplings that accounts for cloud cover and for a positioning of the satellite sub-tracks that is not necessarily located over the Paris area. In order to separate it from the impact of systematic errors, an ensemble of inversions is conducted as described above

but ignoring systematic errors, i.e., by defining $\mathbf{e}_y[i]$ as equal to $\mathbf{e}_y^r[i]$ and by deriving the matrix $\mathbf{R}[i]$ as $\mathrm{diag}(\boldsymbol{\sigma}_y^r[i]^2)$. When ignoring the systematic errors, the analysis of the theoretical uncertainty reductions from $\mathbf{B}$ to the set of $\mathbf{A}[i]$, in comparison to those obtained with the more theoretical ("TH") sampling, should illustrate the specific impact of using realistic observation samplings, while the comparison between the set of $\mathbf{A}[i]$ and $\mathbf{A}_{\mathrm{pract}}$ should help characterizing the impact of the Monte Carlo sampling errors. Furthermore, the results from the Monte Carlo experiments are analysed in terms of

comparison of the practical posterior uncertainties, $\mathbf{s}_{\mathrm{pract}}^{\mathrm{bias}}$ and $\mathbf{A}_{\mathrm{pract}}$, to practical prior uncertainties, $\mathbf{s}_{\mathrm{pract}}^{\mathrm{priorbias}}$ and $\mathbf{B}_{\mathrm{pract}}$, which differ from $\mathbf{0}$ and $\mathbf{B}$ due to the Monte Carlo sampling errors.

### 2.8.4 Monte Carlo sampling of the posterior uncertainties when combining the different types of observation errors

In the fourth set of OSSEs, the observation errors combine the biases in the observation operator and the non-Gaussian and biased distribution of the measurement errors considered in the second and third set of OSSEs. This, again, is only tested for

the most realistic spatial sampling of the satellite imagery corresponding to the CarbonSat configuration (SIM-CarbonSat). Accounting for this combination of errors leads to a Monte Carlo ensemble approach solving for, for each ensemble member:

$$\mathbf{e}_s^a[i] = \mathbf{e}_s^b[i] + \mathbf{K}[i]\big(\mathbf{y}^{\mathrm{bias}}[i] + \mathbf{e}_y[i] - \mathbf{M}[i]\,\mathbf{e}_s^b[i]\big) \tag{12}$$

for $i \in [1..N]$ where $\mathbf{K}[i] = \mathbf{A}[i]\,\mathbf{M}[i]^T\,\mathbf{R}[i]^{-1}$ and $\mathbf{A}[i] = \big[\mathbf{B}^{-1} + \mathbf{M}[i]^T\mathbf{R}[i]^{-1}\mathbf{M}[i]\big]^{-1}$

where all the terms are defined and derived similarly as in equation 11 except $\mathbf{y}^{\mathrm{bias}}[i]$ which is the resampling of the $\mathbf{y}^{\mathrm{bias}}$ term described for the second set of OSSEs on the $\mathbf{y}[i]$ structure. Here again, the practical estimates of the posterior uncertainties based on these ensembles need to be compared to the practical prior uncertainties obtained with the corresponding sampling of $N(\mathbf{0},\mathbf{B})$.





### 2.9 Potential sources of uncertainty that are ignored in the OSSE framework

From the most optimistic set-up to the less optimistic one, the OSSE framework accounts for an increasing number of sources of uncertainty in addition to those directly controlled by the inversion, i.e., the temporal or sectorial distribution of the emissions and their 6-hour budget. The measurement errors are fully or partially accounted for in the set-up of $\mathbf{R}$, and

fully accounted for in the inputs of the less optimistic inversion experiments. We investigate the impact of uncertainties in the spatial distribution of the emissions, that of uncertainties in the temporal profiles of these emissions within the Paris area and that of uncertainties in the boundary conditions. The impact of uncertainties in the initial conditions is evaluated along with that of the lateral and top boundary conditions using a coarse resolution product, which will miss the fine scale patterns due to the Paris emissions before the 6-hour period of inversion. The fact that the satellite imagery is not sensitive to these

emissions is not systematic. Therefore uncertainties in these emissions could sometimes have some impact, when the wind speed is particularly low, but it is expected to be negligible.

However, some major sources of uncertainties are ignored even in the less optimistic case, which prevents from considering this case as "fully pessimistic". Above all, atmospheric transport modelling errors are ignored while it can easily generate large uncertainties. If forced with erroneous wind modelling, the simulation of the $XCO_2$ plume from Paris can have wrong

shape and location while the inversion strongly relies on these simulated shape and location to filter the $XCO_2$ signal associated with the emissions of Paris from the images. Despite the rather smooth variations in space of the NEE, uncertainties in the spatial distribution of the NEE may perturb the results since the inversion relies on the patterns from the assumed distribution to separate it from the Paris emissions. Furthermore, this study uses a simulation of the NEE fluxes for October, when these fluxes are rather weak, which, in principle, should yield better results for the FF emissions than if

leading the test in spring or summer when the amplitude of NEE during daytime is much larger. Besides, uncertainties in the anthropogenic emissions outside the Paris area but within the model domain could have a critical impact since sources, e.g., in North Eastern France can be high and their emission plumes could overlap that of the Paris area. Finally, the impact of the biased and non-Gaussian nature of uncertainties in the fluxes is not assessed. However, it is out of the scope of this study to address such a level of issues.

## 3 Results with optimistic configurations of the OSSEs

This section presents the results of the inversions when assuming that the theoretical set-up of the inversion system is fully consistent with the actual errors, i.e., the results from experiments "TH" (TH-2, TH-x-ns, THsect-2, THsect-x-ns with x =4, 6, 8 or 10 and ns = 1 or 2) defined by table 2.



### 3.1 Some preliminary insights from the model simulation of $XCO_2$

For each of the 20 inversion days in October 2010, simulations of the $XCO_2$ field at 11:00 are generated with the observation operator described in section 2.5 and setting $\boldsymbol{\lambda}^b = [1,1\ldots 1]_{n_\lambda}{}^T$ and $b=b^{LMDZ}$, i.e., by imposing the flux fields and the boundary conditions from Airparif, C-TESSEL and the LMDZ simulation without any rescaling or offset of these products.

Figure 1 gives the corresponding image when sampling $XCO_2$ over the full CHIMERE domain at 2 km resolution for one of the inversion days in October.

The analysis of such images and of the response functions computed as described in section 2.5.4 shows that the typical signature of the Paris emissions in the $XCO_2$ field at 11:00 is a plume whose amplitude is ~1 ppm and whose width is ~40 km at a distance of ~150 km downwind from Paris. This amplitude is similar to the typical measurement random noise

assumed for CarbonSat or for the optimistic configuration of the TH-LargeSwath sampling. The spatial gradients of $XCO_2$ that are generated by either the NEE or the boundary conditions have an amplitude that sometimes reaches the same order of magnitude. Stronger (weaker) wind speeds induce narrower, longer and less intense (wider, shorter and more intense) plumes. For strong wind speeds (above 16 ms$^{-1}$ at 700 magl), the $CO_2$ emitted between 5:00 and 6:00 can reach the domain boundaries before 11:00 and does not systematically have a signature in the domain at 11:00. Wind speeds in October 2010

and at 700 magl range quite uniformly between 1 and 17 ms$^{-1}$, which is representative of the range of wind speeds for the full year.

All these indications imply that the wind should be a critical driver of the performance of the inversion. In the following, the results from the experiments for the 20 different inversion days will thus be presented together as a function of the wind speed. This wind speed is characterized hereafter by its values at 700 magl over Paris, assuming that they are well

representative of the transport within the mixing layer during the time window of interest. They are averaged in time over the period between the emission time and the CarbonSat overpass time when inverting hourly emissions, or over the hour before the CarbonSat overpass when inverting sectorial emissions.

### 3.2 Posterior uncertainty when controlling hourly fluxes

Figure 2 displays prior and posterior uncertainties from experiments TH-2 and TH-x-ns when inverting the hourly FF $CO_2$

emissions and the hourly NEE. This section first focuses on results from TH-2. Figure 2a indicates that, for the satellite configuration close to that of CarbonSat, larger wind speeds lead to smaller uncertainty reduction for the 6-hour mean FF emissions. This can be explained by the fact that, with stronger wind, the amplitude of the plume from the city emissions is smaller while the measurement error is independent from the wind speed. Therefore the signal from the emissions is filtered with a smaller signal to noise ratio and thus a higher uncertainty.

However, figure 2a also indicates that, while the rule applies for hourly emissions when wind speeds are higher than ~6 ms$^{-1}$, the uncertainty in hourly emissions increases with weaker wind speed for wind speeds that are smaller than ~6 ms$^{-1}$. This is interpreted as the consequence of the overlap between the signatures of consecutive hourly emissions, and thus to the



decrease of the potential of the inversion to separate these signatures, when the wind speed is low. This is demonstrated by figure 2b, which shows negative correlations between posterior uncertainties in consecutive hourly emissions, whose absolutes values increase when the wind speed decreases. If correlations in the prior uncertainty are all positive or null, which is the case here, negative correlations between posterior uncertainties in control variables arise when the signature of

these control variables in the observation space are not perfectly separated by the inversion system. It reflects some aliasing in the corrections from the inversion with an overestimation of some variables balanced by an underestimation of other variables. The combination of larger standard deviations of the posterior uncertainties in control variables with more negative correlations between these posterior uncertainties indicates larger problems of separation. When the wind speed is high, the correlations between posterior uncertainties in hourly emissions are nearly null, potentially due to either/both the

weak control of the hourly emissions or/and to the absence of overlapping between the signatures of consecutive emissions despite atmospheric diffusion. However, the uncertainty reduction for the hourly emissions becomes weaker for 0-4 ms$^{-1}$ wind speeds than for 4-7 ms$^{-1}$ wind speeds. Therefore, the decrease of the negative correlations below -0.4 when the wind speed decreases below 4 ms$^{-1}$ is primarily driven by the increasing difficulty of the inversion to separate a given 2-hour budget of emissions between the 2 corresponding hours rather than by the increase of the control of the hourly emissions.

This issue for solving for the temporal profile of the emissions at the hourly scale does not alter the ability to get a precise estimate for the overall budget of 6-hour emissions.

The values obtained for the posterior uncertainty in figure 2a are often small compared to the prior uncertainty both for 6-hour mean and hourly emissions. Indeed, for wind speeds smaller than 10 ms$^{-1}$, the uncertainty reduction is nearly equal to or larger than 50% for the 6-hour mean emissions or for the emissions between 10:00 and 11:00. For wind speeds larger than

10 ms$^{-1}$, the uncertainty reduction is generally high for the emissions between 10:00 and 11:00 but that for the 6-hour mean emissions can be as low as 20%. The uncertainty reduction for the emissions between 9:00 and 10:00 ranges between 35% and 50%. It is significantly smaller (between 20 and 50% for wind speeds smaller than 7-8 ms$^{-1}$, and less for larger wind speeds) and it has a stronger sensitivity to wind speed for the hourly emissions before 9:00. The signatures of these emissions have been far more diffused through atmospheric transport when observed at 11:00 than that of emissions just

before the satellite overpass. This explains these results since atmospheric diffusion decreases the amplitude of these signatures and increases their overlap. It is shown by the values of correlations of the uncertainties in consecutive hourly emissions in figure 2b, which are generally more negative for earlier hours. The potential of the system for solving for the temporal profiles before 9:00 is thus rather weak. The uncertainty reduction for the emissions before 7:00 is generally null for wind speeds larger than 11 ms$^{-1}$ due to the signature of the emissions exiting the TH-CarbonSat sampling area.

Hereafter in the text and in the figures, the prior and posterior uncertainties in the fluxes are quantified in a relative way since the inversion system controls scaling factors for the flux estimates in the observation operator, from Airparif and C-TESSEL. For simplification, we will speak about "relative prior and posterior uncertainties", but one should thus keep in mind that these uncertainties are relative to the estimates of the fluxes in the observation operator, not to the prior or posterior estimates of the fluxes. The relative posterior uncertainty in 6-hour mean emissions ranges from 5% to 17% in



experiment TH-2. This large spread of the results highlights the critical role played by wind speed in defining the potential of the assimilation of the satellite image for monitoring the Paris emissions.

The correlations between the posterior uncertainties in NEE and $FFCO_2$ emissions are nearly null (not shown). The rather high uncertainty reduction for the Paris FF emissions obtained with experiments TH-2 can thus be partly explained by the

good separation between the signature of these emissions and that of the NEE despite the spatial overlapping of these signatures. However, the uncertainty reduction for NEE (not shown) is actually much smaller than for the $FFCO_2$ emissions due to the lower amplitude of their signature (in October). The problem of separating between $FFCO_2$ emissions and NEE may thus be eased by the choice of conducting experiments in October when the NEE and thus the prior uncertainty in NEE is relatively small. Higher negative correlations between posterior uncertainties in $b$ and $FFCO_2$ emissions (not shown) can

be explained by the larger amplitude of prior uncertainties in $b$ than in the NEE. It does not prevent the system from getting a large uncertainty reduction for FF emissions. But a significant part of the posterior uncertainty in FF emissions can be connected to the uncertainties in fluxes outside the model spatial domain or temporal window that are characterized by prior uncertainties in $b$.

Results when using coarser resolution but wider swath sampling for the $XCO_2$ observation (experiments TH-x-ns; see the

results for TH-4-2 in figure 2c) show a behaviour that is similar to that obtained with TH-2. The uncertainty reduction is smaller than for TH-2 except when considering hourly emissions before 8:00, since the signature of these emissions is often visible in the full satellite field-of-view (FOV) in TH-x-ns while it is partially or fully outside the satellite spatial sampling in TH-2 (see the comparison between figure 2a and figure 2d). The skill in decreasing the uncertainty for hourly emissions after 8:00 or for wind speeds smaller than 10 ms$^{-1}$ is strongly limited by the observation of $XCO_2$ at a resolution of 4 km or more

with a precision of 1.2 ppm or poorer compared to that when observing $XCO_2$ at 2 km resolution with a precision of 1.1 ppm. A similar limitation occurs for the reduction of the uncertainty in 6-hour mean emissions.

Figure 2d illustrates the dependence of the uncertainty reduction for 6-hour mean emissions to the changes in the satellite measurement configuration, and mainly to the changes in the spatial resolution and precision of the observation. The relative differences of the scores of uncertainty reduction from experiments TH-x-ns compared to the ones from TH-2 are influenced

by the wind speed. In particular, for high wind speeds, the uncertainty reduction is slightly larger for TH-4-2 than for TH-2 since the former gets the signature of the emissions before 8:00. However, for wind speeds smaller than 12 ms$^{-1}$, the use of a larger swath for the observation does not impact the mean results significantly, and the uncertainty reduction is approximately decreased by half when dividing the spatial resolution by two or when multiplying the measurement noise by two. Consequently, even if considering results for the lowest wind speeds only, in order to expect a 50% uncertainty

reduction in the 6-hour mean emissions, one should not use an imagery with a resolution $\geq$ 6 km and a measurement error close to 2 ppm or poorer.

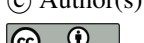



### 3.3 Posterior uncertainty when controlling the fluxes per sector and land cover type

The results of the inversions when controlling the budget of the fluxes per sector of anthropogenic activity or per land cover type are detailed below for experiment THsect-2. As when controlling the hourly variations of the emissions (see section 3.2), the results from experiments THsect-x-ns are qualitatively similar but with higher standard deviations of the posterior uncertainties than in THsect-2.

The results for the 6-hour mean $FFCO_2$ emissions or NEE are very similar to what was seen when controlling the hourly variations of the emissions (see section 3.2). Accounting for uncertainties in the temporal profiles or in the distribution per type of underlying processes does not modify the ability to get the full budget of the 6-hour fluxes. In particular, it does not significantly modify the ability to separate the signature from the $FFCO_2$ emissions and that of the NEE or that of $b$.

However, the reduction of uncertainty for the individual sectors of emissions shown in figure 3a is rather weak. The strong negative correlations between the uncertainties in the different sectors of FF $CO_2$ emissions in figure 3b confirm that the system hardly separates the total budget of the emissions between the different sectors due to the overlapping of their signatures in the $XCO_2$ field. This is particularly true when considering the highest emissions sectors i.e. transport and residential combustion. The spatial and temporal distribution of the different sectors of emissions cannot prevent such an overlapping since all sectors are distributed all over the Paris urban area.

The uncertainty reductions hardly exceed 25% for a given sector of emission. The best results are not obtained for the most emitting sector (road transport and residential combustion) but for the sector whose spatial distribution differs the most to that of the other sectors, i.e., industrial combustion. They can reach 50% for low wind speed but such a value is not representative of the uncertainty reductions obtained for wind speeds smaller than 10 ms$^{-1}$, which rather range between 13% and 30%.

Overall, these results indicate that, without additional information, the $XCO_2$ satellite imagery cannot distinguish between the various $CO_2$ emitting sectors of the Paris area, but that the uncertainty in the sectorial distribution of the emissions does not impact the skill for monitoring the total emissions. Therefore, in the following, we do not analyse the impact of additional sources of uncertainties when controlling this sectorial distribution of the emissions but only when controlling their temporal profile.

## 4 Results with less optimistic configurations of the OSSEs

### 4.1 Impact of biases in the observation operator

#### 4.1.1 Biases in the spatial distribution of the Paris emissions

The biases in the posterior estimate of the scaling factors for the emissions of Paris due to the biases in the spatial
30 distribution of the emissions are illustrated for representative cases of the Bdist-2-r and Bdist-x-ns-r set of OSSEs: Bdist-2-15 in figure 4a, Bdist-2-45 in figure 4c and Bdist-8-2-45 in figure 4e. For these cases, the spatial distribution of the



emissions is described in the observation operator as a homogeneous distribution in a disk ("15" and "45" referring to the radius of the disk, in km), while the true distribution is assigned to be that of the Airparif inventory. Figures 4b, 4d and 4f show the corresponding updates of the posterior uncertainties due to the change of observation operator.

The biases for Bdist-2-15 are negative. They can reach very high negative values: -20% to -40% for the 6-hour mean
emissions, below -50% for the hourly emissions. They are generally below -15% for both hourly and 6-hour mean emissions. The root sum square of such biases and of the random posterior uncertainties shown in figure 4b can be considered as the total uncertainties in the emissions. Since the biases are higher in terms of absolute value than the prior uncertainties, they imply an increase rather than a decrease of the total uncertainties from the inversion. These biases are smaller but still very large for high wind speeds.

The biases for Bdist-2-45 are qualitatively and quantitatively very different from that for Bdist-2-15. They are systematically positive for the 6-hour mean emissions and positive most of the time for the hourly emissions, but they can also be negative. The amplitude of the biases decreases with higher wind speeds. The values are smaller than for Bdist-2-15 and do not exceed, in terms of absolute values, 15% for the 6-hour mean emissions. They hardly exceed 20% for the hourly emissions. However, the reduction of the random component of the uncertainty in hourly emissions is generally much lower for Bdist-
2-45 than for Bdist-2-15. The scores of uncertainty reduction obtained with TH-2 are between those for Bdist-2-15 and those for Bdist-2-45 and all these experiments show the same qualitative dependency of the results to the wind speed or to the time of the hourly emission considered.

The negative biases with Bdist-2-15 can be explained by the fact that the response functions computed by the inversion system have a smaller extent and a larger amplitude than the "true" emission signatures in this OSSE since the spatial extent
of the emissions of Paris in the observation operator is smaller than the actual one. In the extent of the response function of the city emissions modelled with the observation operator, $\left(\mathbf{M}_{\text{inventory}}^{\text{true}} - \mathbf{M}_{\text{inventory}}\right)\mathbf{s}^{\text{true}}$ is thus generally negative and it converts into a negative bias in the control vector of the city emissions. A practical interpretation of this is that the system uses only one part of the observed plume of $XCO_2$ to compute the budget of $XCO_2$ due to the emissions from Paris, and thus the budget of these emissions, which negatively biases the results. Conversely, in experiment Bdist-2-45, the modelled
extent of the Paris emissions is larger and has smaller amplitude than the actual one. The response function to the modelled emissions fully covers the actual plume of $XCO_2$ from the Paris area. Therefore the inversion system does not miss any portion of the observed $XCO_2$ plume when computing the budget of $XCO_2$ due to the emissions from Paris. This explains why the resulting bias in the emission budget is relatively small. Still, the term $\left(\mathbf{M}_{\text{inventory}}^{\text{true}} - \mathbf{M}_{\text{inventory}}\right)\mathbf{s}^{\text{true}}$ is generally positive within the area of the actual plume and generally negative in the area of the modelled plume that is not covered by
the actual plume and the inversion is likely mostly driven by the core of the plume where the NEE signature is, in relative, smaller. This may explain that the bias is positive.

The use of a spatial extent of the emissions that is larger in the observation operator than in the truth implies that the inversion system assimilates more measurement noise in the area covered by the emission response functions and has issues



with a larger overlapping of the signatures of the $FFCO_2$ and of the NEE. This explains the higher posterior uncertainties obtained with Bdist-2-45 or TH-2 than with TH-2 or Bdist-2-15 respectively. Assuming a larger area for the $FFCO_2$ emissions in the observation operator also results in having a larger overlapping of the different response functions to hourly emissions, which hampers the uncertainty reduction for hourly emissions. This can explain that the sensitivity to the changes

of the spatial distribution of the emissions in the observation operator is weaker for the posterior uncertainty in 6-hour mean emissions than for the posterior uncertainty in hourly emissions.

Consequently, using an upper bound for the extent of the emission distribution when this distribution is poorly known appears to be a conservative decision for the inversion. It avoids taking the risk of missing some parts of the urban area with significant emissions, which would bias the estimates. However, there is therefore a compromise to find between this need to

avoid biases and the need for keeping a high uncertainty reduction, which requires not spreading the emissions too much in the observation operator.

The biases obtained with Bdist-8-2-45 (figure 4d) are far smaller than those obtained with Bdist-2-45, while, as already analysed in section 3.2, the posterior random uncertainties from Bdist-8-2-45 (figure 4e) are much larger than those obtained with Bdist-2-45. This is due to the fact that the corrections applied by the inversion system to the prior estimate of the

control parameters when using a lower resolution and noisier imagery (i.e. less and noisier observations) are smaller. This translates into the computation of a "smaller" gain $\mathbf{K}$, which yields smaller uncertainty reduction but also a smaller sensitivity to biases in the observation error. Results from other Bdist-x-ns-r experiments follow the behaviour analysed here and in section 3.2, i.e., they show that larger r, larger x or smaller ns lead to smaller uncertainty reduction but smaller biases. These results indicate the need for having good knowledge on the spatial extent of the Paris urban emissions before

assimilating the satellite imagery, but that knowledge of their distribution within the emitting area need not be precise. This corroborates the results analysed in section 3.3 regarding the weak sensitivity of the inversion to the spatial or sectorial distribution of the emissions within the Paris emitting area. Of note is that this conclusion may be driven by the specificities of Paris in terms of spatial extent and distribution of the emissions.

### 4.1.2 Biases in the boundary conditions

The biases on the estimates of $FFCO_2$ emissions from Paris due to the biases on the transport model $CO_2$ boundary conditions that are described in section 2.8.2 are analysed with experiments Bbc-2 (figure 5a) and Bbc-x-ns; only the results for experiments Bbc-2 and Bbc-8-2 are shown. The biases in both 6-hour mean or hourly emissions can be very high (up to +/-60%) and are generally comprised between ±10% for Bbc-2. There is no clear correlation of the biases with the wind speed even though the gain $\mathbf{K}$ of the inversion is smaller for high wind speeds. The impact of biases in the boundary

conditions depends on the spatial structure of their signature in the observation space. The largest biases in the posterior emissions are obtained when the spatial structures of these signatures overlap with the $XCO_2$ plume from Paris. As when comparing Bdist-x-ns-r experiments to Bdist-2-r experiments in section 4.1.1 and, again, because when using a lower



resolution and noisier imagery the corrections and thus the sensitivity to biases in the observation errors are smaller, the biases obtained with Bbc-x-ns experiments are smaller than but highly correlated to that obtained with experiment Bbc-2.

### 4.2 Impact of the use of a realistic distribution of measurement sampling and errors

The following focuses on the third and fourth sets of OSSEs. They are based on Monte Carlo frameworks using realistic CarbonSat-like samplings of $XCO_2$ which account for cloud cover and a realistic simulation of the satellite FOV, and using a realistic simulation of both the standard deviation of the measurement noise and of the systematic measurement errors. The MC-2 experiment (figures 6, 7a and 7b) assesses the impact of the limitation of the observation sampling due to cloud cover and instrument FOV, of the Monte Carlo sampling errors, and, to a lesser extent, of the use of realistic simulations of the measurement noise. Through its comparison to the MC-2 experiment, the MCsyst-2 experiment assesses the impact of the systematic errors (figures 7c and d).

Figure 6 shows the theoretical uncertainty reduction for the 6-hour mean emissions in MC-2 inversions obtained with the 1[st], 5[th], 10[th], 15[th], 19[th] and 25[th] best observation sampling provided by the simulation of Buchwitz et al. (2013) for each inversion day (see section 2.8.3). The best observation sampling corresponds to a full coverage of the area within a distance of 100 km from the centre of Paris (figure S1) and within an even longer distance in the direction of the emission plume, which depends on the inversion day. This explains why, for a given inversion day, this best observation sampling yields a theoretical uncertainty reduction in experiment MC-2 that is comparable to that of the experiment TH-2 (figure 2) with an optimistic sampling of the area over a 150 km distance from the centre of Paris without cloud cover (TH-CarbonSat). Therefore, in a general way, figure 6 illustrates the strong impact of the cloud cover and of the limitations associated with realistic instrument FOV through the rapid decline of the theoretical uncertainty reductions between the use of the best observation sampling and that of the following ones. In particular, this figure illustrates the fact that, for the inversion days when the wind speed is above 5 ms[-1], the theoretical uncertainty reduction when using the 19[th] best observation sampling is systematically lower than 30%. It is even lower than 20% in most cases. This is generally twice as low as the uncertainty reduction obtained with the 5[th] to 10[th] best observation samplings. The uncertainty reduction rapidly decreases to nearly null values when using the 20[th] and following "best" observation samplings. This reveals that the large majority of the observation samplings simulated throughout the year 2010 brings a weak theoretical constraint on the emission inversions. Even though they do not correspond to the 19 best samplings of any inversion day, the 19 sampling cases of Figure S1 illustrate how the spatial coverage of the Paris area by the images decreases due to cloud cover and to shifts of the satellite FOV away from the centre of Paris. This figure shows that only 18 samplings provide observations for more than 15% of the area within a 100 km distance from the centre of Paris. The cloud cover removes 35% to 70% of the potential data in the satellite field of view near Paris for half of the samplings shown in this figure.

The mathematical framework of the MC-2 and MCsyst-2 experiments is such that, statistically, the limitation of the observation samplings does not bias the practical inversion estimates by itself. However, the Monte Carlo estimations are based on ensembles of 19 members only, which can generate significant Monte Carlo sampling errors. In particular, the





mean of these ensembles can be significantly different from zero. The Monte Carlo sampling errors and biases in the posterior ensembles can be influenced by the specific coupling of the different simulations of cloud cover with the different samples of the prior uncertainty and of the measurement noise when building the Monte Carlo ensembles. Figure 7a shows that these biases hardly exceed +/-5% for both the prior and posterior estimates of the 6-hour mean emissions. Posterior

biases in the 6-hour mean emissions are not significantly larger than prior biases even though they bear additional biases from the subsampling in the observation space. However these biases can sometimes exceed +/-15% for posterior estimates of the hourly emissions.

The impact of Monte Carlo sampling errors for the random uncertainty reduction is characterized by the differences, in experiment MC-2, between the practical uncertainty reduction (figure 7b) and the theoretical uncertainty reduction for the

individual spatial samplings used in this ensemble experiment (figure 6). The practical estimates of the uncertainty reduction for 6-hour mean fluxes have a negative correlation to the wind speed as the theoretical uncertainty reduction. Ranging from -4% to 70%, they tend to correspond to the theoretical uncertainty reductions when using the $10^{th}$ out of 19 best spatial samplings, which range from 10% to 70%. This indicates that the Monte Carlo sampling error has a limited impact on the practical computation of the random uncertainty reduction. Therefore, the Monte Carlo framework appears to be appropriate

for exploring the impact of systematic errors. However, some specific values of the practical uncertainty reduction for the 6-hour mean emissions, e.g. 60% for a wind of ~8 ms$^{-1}$ or the negative value for a wind of ~17 ms$^{-1}$ (figure 7b), reveal significant Monte Carlo sampling errors. The impact of the Monte Carlo sampling errors are even more highlighted by the significant number of negative practical uncertainty reductions for hourly emissions (figure 7b) while, in theory, given the statistics of errors in experiment MC-2 and the assumption that they are perfectly accounted for by the inversion system, the

uncertainty should be decreased by the assimilation of satellite data.

The impact of the systematic errors is given by the direct comparison of figures 7c and 7d to figures 7a and 7b respectively. There is a significant impact of the systematic errors in the posterior biases for both the 6-hour and 1-hour mean emissions (figures 7a and c). While posterior biases in 6-hour mean emissions ranged quite evenly between ±5% in MC-2 such as the prior biases, the posterior biases are shifted towards -5% in MCsyst-2. Similarly, while posterior biases in hourly emissions

are generally comprised between ±10% in MC-2, their spread is larger and shifted to negative values and they often reach -20% in MCsyst-2. The shift towards negative values can be explained by the fact that the systematic errors as modelled by Buchwitz et al. (2013) are biased negatively in the Paris area. This shift between the practical biases from MC-2 and MCsyst-2 indicates that systematic errors could typically generate 3-4% biases in the 6-hour mean emissions. The biases for the posterior estimates of the emissions between 10:00 and 11:00 in MCsyst-2 are nearly systematically negative, unlike that

for other hourly emissions. There is a single positive value while other values are comprised between -5% and -20%. This is due to the urban albedo in the Paris area, which generates specific systematic errors over this area, i.e. where the signature of the emissions between 10:00 and 11:00 stand at the time of the satellite overpass.

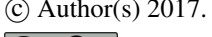



The uncertainty reductions shown in figure 7d are much smaller than those shown in figure 7b. The random uncertainty reductions for 6-hour mean emissions drop from 55%-70% to ~55% for low (< 3 ms$^{-1}$) wind speeds and from 15%-60% to 5%-35% for 3 to 15 ms$^{-1}$ wind speeds. The number of negative uncertainty reductions for hourly emissions is not necessarily larger in MCsyst-2 than in MC-2, but the practical uncertainty reductions for hourly emissions are also strongly decreased

when accounting for the systematic errors, due to the fact that the distribution of these errors is not Gaussian. The combination between the limited practical random uncertainty reductions and the biases due to systematic errors in experiment MCsyst-2 sometimes implies an increase of the total uncertainties because of the inversion, and in a general way a strong decrease of the total uncertainty reductions compared to results in experiment MC-2.

**4.3 Combination of operator biases and of realistic measurement sampling and errors**

Figure 8 illustrates the impact of combining the two types of operator biases analysed in section 4.1 and the realistic cloud cover, observation sampling and random and systematic measurement errors in the Monte Carlo experiment ALL-2. The test is conducted with the use of $\mathbf{M}_{45}^{\mathrm{disc}}$ in the observation operator while the observations are generated using the distribution of the emissions from Airparif. There is no test using $\mathbf{M}_{15}^{\mathrm{disc}}$ in the observation operator but the analysis in section 4.1.1 indicates that such a test would lead to a much larger combination of posterior biases and random uncertainties.

Due to the Monte Carlo framework where the cloud cover and satellite FOV vary from one ensemble member to the other one, the impact of biases in the spatial distribution of the emissions or in the model $CO_2$ boundary conditions of the observation operator is "randomized" in the sense that it varies from one member of the ensemble to the other one. Consequently, adding these biases in ALL-2 compared to MCsyst-2 both increases the biases to very large values in the posterior estimates of the emissions (compare figure 8a and 7c) and dramatically degrades the random uncertainty reductions

down to large negative values. A part of this decrease should also be attributed to the decrease of the random uncertainty reductions when using $\mathbf{M}_{45}^{\mathrm{disc}}$ instead of the maps from the Airparif inventory in the observation operator (see section 4.1.1) but such a decrease is relatively small compared to that seen between experiments MCsyst-2 and ALL-2. In experiment ALL-2, accounting for the biases and random uncertainties, the total uncertainties in the posterior estimates of the emissions are generally larger than the total prior uncertainties.

**5 Discussion and conclusions**

This study investigates the potential of satellite $XCO_2$ spectro-imagery for the atmospheric inversion of the FFCO$_2$ emissions from a megacity. More precisely, it investigates the potential of 2 to 10 km resolution $XCO_2$ spectro-imagery from sun-synchronous missions based on SWIR absorption measurements. Examples of plans for such missions are the CarbonSat mission, which was a candidate to ESA's Earth Explorer 8 opportunity, and the European mission currently studied by ESA

and the European Commission in the context of an increasing political interest for a space borne monitoring of the $CO_2$ anthropogenic emissions (Ciais et al, 2015). However, it also gives some general insights on the potential of SWIR spectro-



imagery from other types of $CO_2$ missions e.g. the GeoCARB geostationary mission, which has been selected as an Earth Venture Mission by NASA. The study focuses on a relatively easy test case: the monitoring of the emissions from the Paris urban area, whose annual budget is approximately 11-14 MtC.y$^{-1}$. The assessment of the imagery potential is based on an analytical inversion system and on an OSSE framework developed specifically for this study. They incorporate different

levels of assumptions, from the traditional and rather optimistic assumptions underlying atmospheric inversion activities to less idealistic ones. The potential of the imagery is quantified in terms of reduction of uncertainty in the emission estimate, i.e. the relative difference between the uncertainty in the flux prior knowledge exploited by the inversion and that in the inverted (posterior) estimate of the emissions.

An image of the $XCO_2$ plume from the Paris urban area at a given time does not bear a significant signature of the Paris

emissions more than 6 hours before. Since, the CarbonSat mission was expected to overfly the Paris area in the morning around 11:00 local time, the inversion framework consists in retrieving the emissions between 5:00 and 11:00 based on an image at 11:00 for a given day. The wind conditions influence the shape and amplitude of the $XCO_2$ plume from Paris, and thus the imagery potential, which is higher when the emitted $CO_2$ signal is larger. The inversions are thus conducted for 20 different days of October 2010, which are representative of the range of wind conditions in the Paris area throughout the

year, and the sensitivity of the results to the wind speed is analysed.

The realistic simulation of the CarbonSat sampling accounting for cloud cover from Buchwitz et al. (2013) indicates that there should be typically 20 days per year during which the satellite could deliver "appropriate" images that could be used to provide a significant theoretical uncertainty reduction on the emission estimates. Satellites with far larger swath (typically 2000 km for Sentinel-5 instead of ~240 km for CarbonSat) and geostationary missions like GeoCARB should provide

"appropriate" images at a higher frequency than CarbonSat. Furthermore, the present study does not analyse the potential information brought by the combination of multiple views by the frequent revisit from a given satellite or through the combination of the data from different missions, and it focuses on the potential of individual satellite overpasses. Nevertheless, the low number of "appropriate" acquisitions from CarbonSat within one year reminds that a large fraction of the days during the year are inappropriate to satellite observation of Paris in the SWIR as a result of cloud cover. This, and

the fact that the satellite imagery does not bring direct information on the emissions earlier than ~6 hours before the overpass demonstrate that the satellite SWIR measurement based $CO_2$ observation alone cannot provide a valid estimate of the daily to annual emissions. In particular, the satellite imagery from a sun-synchronous satellite does not bring direct information on afternoon and night-time emissions if the satellite overpass is around noon. Geostationary SWIR measurements, e.g., with the GeoCARB mission, can fill a large part of this gap in the diurnal cycle of the emissions by potentially providing SWIR

data in the morning and in the afternoon and several measurements over a given urban area per day. However, they cannot cover a significant part of the night. Ancillary information such as ground based measurements, other types of satellite measurements or accurate "bottom-up" knowledge of the temporal profiles from the inventories are required for such purpose. Obtaining such information for a large number of diverse cities may be challenging. Without ancillary information, the satellite $CO_2$ spectro-imagery alone may still be useful to monitor trends. This perspective is particularly interesting



when the trends of the monitored emissions, i.e., the morning emissions in the case of close-to-noon imagery, are representative of the whole diurnal cycle.

The potential and limitations of different levels of image resolution and precision for monitoring the 6-hour emissions before the satellite overpass are investigated with the rather optimistic configuration of the inversions. In this case, the only sources

of errors that are accounted for are uncertainties in the 6-hour mean budgets of the emissions from Paris and of the NEE in Northern France, uncertainties in the hourly or sectorial/per ecosystem distribution of the emissions from Paris and NEE, uncertainties in the $XCO_2$ background and satellite measurement noise. Further, the limitation of the observation sampling due to cloud cover or to the fact that the satellite swath is not centred over Paris are ignored. These experiments reveal that one should not use an imagery with a resolution $\geq 6$ km and a measurement error close to or larger than 2 ppm if targeting a

50% uncertainty reduction in the 6-hour mean emissions from a ~22% relative uncertainty in the prior knowledge on these emissions. When using an imagery with a 2 to 4 km resolution and a 1.1 to 1.2 ppm precision, a 5% to 10% precision on the estimates of the 6-hour mean emission can be reached for favourable atmospheric conditions, i.e. when the wind speed in the boundary layer is less than 9 ms$^{-1}$. However, the uncertainty reduction would not be better than 25% if wind speeds > 15 ms$^{-1}$, even with a 2 km resolution and a 1.1 ppm precision for the measurements. The advantage of using coarser resolution but

larger swath imagery for increasing the frequency of revisit over a given city is not investigated in this study, which focuses on the potential of a mission for days when an image is available. Multi-day inversion frameworks would be required to find the best compromise between the resolution and swath of the imagery.

The optimistic experiments also indicate that the imagery could provide information on the temporal profile of the emissions, even though the corresponding estimates of uncertainty reductions for hourly emissions are significantly smaller

than those for the 6-hour mean emissions. Conversely, the OSSEs indicate a poor ability to solve for the sectorial distribution of the emissions of Paris, and likely their spatial distribution within a known extent. Because of atmospheric diffusion, the differences between the spatial structures of the various sectorial emissions are not large enough to allow for the distinction of their signatures in the $XCO_2$ images. While the residential and traffic sectors are spread throughout the Paris urban area, the commercial and industrial sectors are more localized, which explains why the uncertainty reduction is slightly higher for

these sectors. The lack of ability for monitoring the sectorial distribution of the emission is thus likely strongly related to the specificities of the spatial distribution of the emissions in Paris, which are dominated by diffuse and spatially mixed sources. Less optimistic experiments are then conducted. They investigate the impact of the uncertainties in the spatial extent of the emissions of Paris, and in the $XCO_2$ patterns induced by fluxes outside the Northern France modelling domain or before 5:00, i.e., by $CO_2$ initial, lateral and top boundary conditions in the transport model. They also investigate the impact of the

limitation of the observation sampling due to cloud cover and to the distance between Paris and the satellite sub-track. Finally, they investigate the impact of the systematic measurement errors due to current uncertainties in the radiative transfer modelling underlying the conversion of SWIR measurements into $XCO_2$ data. Biases in the knowledge of the spatial extent of the emissions are shown to be responsible for large negative biases in the inverted emissions if the assumed distribution covers an area smaller than the actual emission area. The Paris urban area is thus not seen as a "point source" whose size





would be negligible even if the results from the sectorial inversions indicated that the inversion is weakly sensitive to the distribution of the emissions within the emission area. Spreading the virtual emissions over a large area that necessarily fully covers the actual emission area is a conservative solution for avoiding large negative biases but it implies smaller uncertainty reductions than when using the proper spatial extension. Biases in the transport model $CO_2$ boundary conditions that bear the

signature of remote fluxes or of fluxes before 5:00 can yield very large biases in the inverted emissions. Finally, when accounting for a realistic observation sampling hampered by cloud cover and systematic errors, the experiments indicate significant biases in the inverted emissions, and moderate random uncertainty reduction compared to the idealistic experiments. The quantitative validity of these last results, derived using a Monte Carlo approach, is limited by the Monte Carlo sampling errors associated with the small ensembles used for the computations. They nevertheless indicate that, for

many meteorological and atmospheric conditions, the inversion based on the satellite imagery would hardly improve the prior knowledge on the emissions.

The very last experiments combining all these sources of errors show the worst results with posterior estimates bearing larger uncertainties than the prior ones. One may argue that these OSSEs use pessimistic assumptions regarding the uncertainties in the spatial distribution of the emissions and in the model boundary conditions. They nevertheless strengthen

the conclusion that monitoring the emissions from the Paris urban area with the coupling between state of the art inversion systems and $XCO_2$ observation capabilities is a difficult challenge. It will be even more difficult for cities with smaller emissions than Paris, which is the case of nearly all cities in Europe, and for cities that have other major cities or large combustion plants in their vicinity. The complementary study by Pillai et al. (2016) assessed the potential of CarbonSat for monitoring the $CO_2$ emissions of Berlin. The annual emission budget in Berlin is similar to that in Paris, even though Berlin

is less populated, since this city bears much larger power plant emissions. Pillai et al. (2016) used the same simulations of CarbonSat data sampling and errors from Buchwitz et al. (2013) as here, but different theoretical frameworks both for the statistical characterization of the errors and the inversions, and different configurations for the modelling and the sources of errors. However, they found results for the mean emissions corresponding to a plume seen during an overpass that are quite similar to those here with 5-20% random posterior uncertainties depending on the meteorological conditions when ignoring

systematic errors in the data and errors in the modelling system, and with high impacts of the systematic errors and of biases in the modelling system.

Based on our results, both the monitoring of the city emissions based on good prior knowledge which is investigated here, and the fully independent verification of an inventory of these emissions using a satellite $XCO_2$ imagery appear as long term objectives rather than already mature techniques. This general conclusion is derived based on computations that use both

optimistic and pessimistic assumptions. Section 2.9 lists a number of sources of errors that have been ignored in this study, in particular atmospheric transport modelling errors. Present pixel-based inversion techniques strongly rely on the position and extent of the city $XCO_2$ plume simulated with atmospheric transport models. They assume that transport errors consist in random and unbiased noise with rather simple spatial correlation structures at the image pixel level. In practice, as a consequence of errors in the wind speed and direction, the simulation of the $XCO_2$ plume position and structure can be



poorly related to the actual ones. Large transitory wind direction errors could even lead to situations when the inversion system would not "see" the $XCO_2$ excess due to the city emissions, in which case the pixel-based assimilation technique would yield quasi null emission estimates. An inversion procedure that would control simultaneously both transport parameters and $CO_2$ emissions within a coupled meteorological-$CO_2$ transport model (Kang et al., 2011) may partially solve

these issues. However, due to the complexity of such an approach, it has hardly been investigated in the inverse modelling community, and the result would likely bear significant residual transport errors. The atmospheric transport errors may thus degrade the scores of uncertainty reduction obtained in this study. The NEE during spring and summer that is far larger than in October, and the uncertainties in the anthropogenic emissions outside the Paris area have also been identified as other sources of errors that would degrade these scores of uncertainty reduction.

Conversely, some other components of the experimental framework could be viewed as pessimistic. The way we quantified the impact of uncertainties in the spatial distribution of the emissions and in the transport model $CO_2$ initial, lateral and top boundary conditions may be seen as extreme (see section 2.8.2). Indeed, it is treated as a bias that is fully ignored by the inversion. The inversion approach generally use a random representation for all types of errors, including the transport errors due to wrong meteorological forcing, even though it could be abusive when these errors relate to processes that seem quite

deterministic at the considered timescales. This explains why the cloud cover has been taken as a random parameter of the observation vector in this study. This also explains why systematic errors, which can depend on atmospheric transient properties such as aerosol load, have been considered as random errors. In theory, one could also take this statistical point of view for the errors in the observation operator that have been investigated here i.e. in the spatial distribution of the emissions and in the boundary conditions. Then, the configuration of the observation error in the inversion system could have been

inflated to anticipate for these errors. Still, the complex spatial correlations of the errors from the spatial distribution of the emissions and from the boundary conditions would have hardly been modelled in the inversion system. Therefore, the bias computation in this study can be seen as a qualitative rather than quantitative assessment of the sensitivity to such sources of errors.

It would also seem pessimistic to assume that uncertainties in the emissions spatial distribution could be such that the best

knowledge on this distribution would be a homogeneous distribution over a disk. High resolution satellite imagery of the urban land cover in the visible spectrum from other Earth observation missions should provide at low cost a finer representation of the city, at least in terms of spatial extent, but also regarding the location of power plants, road networks, and industrial, commercial or housing areas. The quantitative impact of uncertainties in the spatial distribution of the emissions could thus be quite smaller than that shown in this study. It does not alter the conclusions regarding the need for a

conservative mapping of the emissions in the observation operator over an area that is identical or larger than the actual area of emissions.

As for most inversion studies, a critical parameter of the experiments that is not objectively defined is the prior uncertainty covariance matrix. Only one configuration of this matrix has been tested in this study with a 50% uncertainty in the hourly emissions or sectorial budgets of the emissions between 5:00 and 11:00. It could sound rather optimistic for the inversion,



since lower prior uncertainties would give lower potential to the inversion for uncertainty reduction and since 50% is a rather high uncertainty for hourly emissions or for the wide sectors analysed in this study, at least for cities like Paris whose emissions have been regularly inventoried. In general, uncertainties in existing "bottom-up" inventories at city scale are extremely difficult to define, but the order of magnitude for the emissions in cities in Europe should be known

approximately even when considering hourly emissions or budgets for the transport, residential, commercial and industrial sectors. However, such a prior uncertainty could be too low for cities where there is a critical lack of data on the fossil fuel consumption, e.g., in developing countries..

Furthermore, the discussion regarding the correlations between prior uncertainties in the different hourly or sectorial emissions is even more difficult, and the resulting ~22% to 26% uncertainties in 6-hour mean emissions is debatable. Indeed,

if a significant fraction of the uncertainty in the emissions derives from the uncertainty in emissions factors used to build the inventories, a large positive correlation is expected between the uncertainties in the various hourly emission estimates. Current inventories provide estimates for average conditions and do not represent transient conditions either related to natural (e.g. cold spell) or human (e.g. holiday, socio-economic events) causes that may apply to several hours in a row. Therefore, large temporal correlations of the uncertainties in hourly emissions or large correlations between uncertainties in

sectorial budgets are expected if a significant fraction of these uncertainties are linked to such transient conditions. However, the usual representation of temporal correlations for prior uncertainties in natural fluxes that uses positive correlations decreasing with increasing time lags are not well suited to anthropogenic emissions, which have complex cycles at daily, weekly and annual scales and a large variability. Note that the existence of negative correlations between prior uncertainties in hourly or sectorial emissions can also be expected when the hourly and sectorial emissions of the inventories are estimated

from the disaggregation of an annual budget. This somewhat justifies that our configuration of the prior uncertainties ignores the correlations between the uncertainties in hourly or sectorial emissions.

Assuming positive correlations would lead to higher uncertainty reductions both because it would increase the prior uncertainty in the 6-hour mean emissions and because it would enable extrapolating the information obtained about emissions just before the satellite overpass to earlier emissions. It would also support the assumption that the satellite

imagery could bear "indirect" information about the emissions earlier than 6-hour before the satellite overpass. From that point of view, the experiments in this study can be seen as pessimistic.

Choosing the Airparif inventory and the CHIMERE transport model as, respectively, the true emissions and transport in the Paris area can bias the diagnostics in this study. Both products are quite diffuse. The Airparif inventory uses typical periodical temporal profiles of the emissions that are homogeneous in space and that ignores transient events over short term

periods, which leads to smooth maps, especially for the traffic emissions. The CHIMERE model is an Eulerian model and, as such, is subject to numerical diffusion. This may be viewed as a limitation for separating different hourly or sectorial components of the emissions, and the overestimation of the spread of the emissions in Airparif could yield smaller scores of random uncertainty reduction than if having the actual distribution of the emissions (see section 4.1.1). However, as long as the observation operators in the inversion systems need to rely on such products, any major discrepancies between such



products and the actual distribution of the emissions or transport would result in additional sources of errors during the inversion. The observation operator for Paris could not rely on a better estimate of the emission distribution than the Airparif inventory, unless investing in the convolution of this inventory with precise statistics of the spatial and time variations of the anthropogenic activities (e.g. precise hourly traffic counting) and, to our knowledge, such a product is not available in any
city presently.

The sensitivity of the inversion to the biases in the transport model $CO_2$ boundary conditions, the negative correlations between the posterior uncertainties in $FFCO_2$ emissions and in the $XCO_2$ background $b$, and the impact of systematic errors demonstrate a rather weak ability of the pixel-based and Gaussian inversion framework to separate the emission plume of Paris from the signatures of other sources or from errors with spatial structures. However, these signatures sometimes seem
easy to distinguish from the emission plume of Paris "by eye" (see figure S2). It is particularly true when considering the uncertainty in $b$. Actually, as discussed in this study, the negative correlations in posterior uncertainties, which characterize such a problem of separation, arise as soon as there is some overlapping between the signatures of the different control parameters. Image processing techniques with pattern recognition could help avoiding such an issue.

Non-Gaussian approaches and account for correlations in the observation errors have already been used in atmospheric
inversions but it generally still relies on simple assumptions. New inversion techniques, potentially based on image data assimilation (Corpetti et al., 2009) could thus help better assimilate $XCO_2$ images. The combination of $XCO_2$ measurements with other types of data may critically improve the inversion technique and its potential. There are high uncertainties and temporal variability in the ratios of emission factors between CO2 and co-emitted species like CO, NOx, COVs and aerosols. However, the joint assimilation of such co-emitted species and $CO_2$ data may help constrain the inversion of $CO_2$
anthropogenic emissions, especially if these data are jointly measured (Rayner et al., 2014). The potential of the spaceborne $XCO_2$ imagery could also be increased by its integration within an observation system with constellation of satellites, ground based networks and airborne instruments. A decrease of the systematic errors associated with the retrieval of $XCO_2$ data from SWIR absorption measurements through an improvement of the inverse radiative transfer models (Buchwitz et al., 2015) would be critical for increasing the potential of the satellite $XCO_2$ imagery. But in parallel, the improvement of
inversion techniques would thus also be needed to increase the potential of such a satellite observation for monitoring the city emissions.

**Acknowledgements.** This study has been co-funded by ESA (LOGOFLUX – CarbonSat Earth Explorer 8 Candidate Mission – Inverse Modelling and Mission Performance Study" - ESA contract no. 400010537/12/NL/CO) and by the Chaire
industrielle BridGES (a joint research program between Thalees Alenia Space, Veolia, the Université de Versailles Saint-Quentin-en-Yvelines, the Commissariat à l'Energie Atomique et aux Energies Renouvelables, and the Centre National de la Recherche Scientifique). We also would like to thank partners of the LOGOFLUX project (in particular NOVELTIS) and of BridGES for the fruitful discussions we had with them regarding this study.





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



| Type of setting | Option | Description |
|---|---|---|
| **Control vector** | Hourly fluxes | $\mathbf{s} = [\lambda_5^{FF}, \lambda_6^{FF}, \lambda_7^{FF}, \lambda_8^{FF}, \lambda_9^{FF}, \lambda_{10}^{FF}, \lambda_5^{NEE}, \lambda_6^{NEE}, \lambda_7^{NEE}, \lambda_8^{NEE}, \lambda_9^{NEE}, \lambda_{10}^{NEE}, b]^T$ |
| | Flux types | $\mathbf{s} = [\lambda_{Transportation}^{FF}, \lambda_{Res.\ Combust.}^{FF}, \lambda_{Comm.\ Combust.}^{FF}, \lambda_{Large\ Indust.}^{FF}$ $\lambda_{Cropland}^{NEE}, \lambda_{Grassland}^{NEE}, \lambda_{NeedleLeaf\ F.}^{NEE}, \lambda_{BroadLeaf\ F.}^{NEE}, b]^T$ |
| **Observation sampling** | TH-CarbonSat | Sampling of the 150 km radius circle centred on Paris at 2 km resolution |
| | TH-LargeSwath | Sampling of the full CHIMERE domain at x km resolution where x=4,6,8 or 10 |
| | SIM-CarbonSat | Sampling from the simulations by Buchwitz et al. (2013) using a 240 km swath |
| **Biases in the observation operator** | Emission map | Using $\mathbf{M}_{inventory} = \mathbf{M}_r^{disc}$ instead of $\mathbf{M}_{inventory}^{true}$ where r = 15 or 45 (in km) |
| | BC | Bias in the Boundary Conditions |
| **Theoretical (system) measurement error** | CS | 1.1 ppm random noise (1 sigma) |
| | Sent5-1 SWIR | 2.1 ppm random noise (1 sigma) |
| | Sent5-2 SWIR | 1.2 ppm random noise (1 sigma) |
| | CS space varying | Simulation of random errors by Buchwitz et al. (2013) |
| | CS incl syst error | Random noise (1 sigma): Root Sum Square of 0.3 ppm and of the simulation of random errors by Buchwitz et al. (2013) |
| **Practical (actual) measurement error** | Theoretical | Consistent with the configuration of the inversion system |
| | Ensemble | Sampling from the simulation of random errors by Buchwitz et al. (2013) |
| | Ensemble incl syst error | Sampling from the simulation of random and systematic errors by Buchwitz et al. (2013) |
| **Relevant diagnostic of the posterior uncertainty** | Theoretical A | Analysis of the posterior uncertainty covariance matrix $\mathbf{A}$ diagnosed by the inversion system (equation 4) |
| | Theoretical A and bias | Analysis of the posterior uncertainty covariance matrix $\mathbf{A}$ diagnosed by the inversion system (equation 4) and of the posterior bias $\mathbf{s}^{bias}$ from equation 7 |
| | Practical A and bias | Analysis of the mean and covariance of the sampling of the posterior uncertainty from the Monte Carlo inversion experiments |

**Table 1. The different options for the configuration of the OSSEs.**




| Name | Control vector | Observation sampling | Biases in the observation operator | Theoretical (system) measurement error | Practical (actual) measurement error | Relevant diagnostic of the posterior uncertainty |
|---|---|---|---|---|---|---|
| **TH-2** | Hourly fluxes | TH-CarbonSat | None | CS | Theoretical | Theoretical A |
| **TH-x-ns** (*) | Hourly fluxes | TH-LargeSwath at x km res | None | Sent5-ns SWIR | Theoretical | Theoretical A |
| **THsect-2** | Flux types | TH-CarbonSat | None | CS | Theoretical | Theoretical A |
| **THsect-x-ns** (*) | Flux types | TH-LargeSwath at x km res | None | Sent5-ns SWIR | Theoretical | Theoretical A |
| **Bdist-2-r** (*) | Hourly fluxes | TH-CarbonSat | Emission map: $\mathbf{M}_r^{disc}$ | CS | Theoretical | Theoretical A and bias |
| **Bdist-x-ns-r** (*) | Hourly fluxes | TH-LargeSwath at x km res | Emission map: $\mathbf{M}_r^{disc}$ | Sent5-ns SWIR | Theoretical | Theoretical A and bias |
| **Bbc-2** | Hourly fluxes | TH-CarbonSat | BC | CS | Theoretical | Theoretical A and bias |
| **Bbc-x-ns** (*) | Hourly fluxes | TH-LargeSwath at x km res | BC | Sent5-ns SWIR | Theoretical | Theoretical A and bias |
| **MC-2** | Hourly fluxes | SIM-CarbonSat | None | CS space varying | Ensemble | Practical A and bias |
| **MCsyst-2** | Hourly fluxes | SIM-CarbonSat | None | CS incl syst error | Ensemble incl syst error | Practical A and bias |
| **ALL-2** | Hourly fluxes | SIM-CarbonSat | Emission map: $\mathbf{M}_{45}^{disc}$ and BC | CS incl syst error | Ensemble incl syst error | Practical A and bias |

**Table 2. The different OSSEs conducted in the study and their configuration, sorting the group of experiments from the most optimistic to the less optimistic ones. (*): rows associated with ensembles of experiments with x designating 4,6,8 or 10, ns=1 or 2 and r=15 or 45. Other notations are defined by Table 1.**



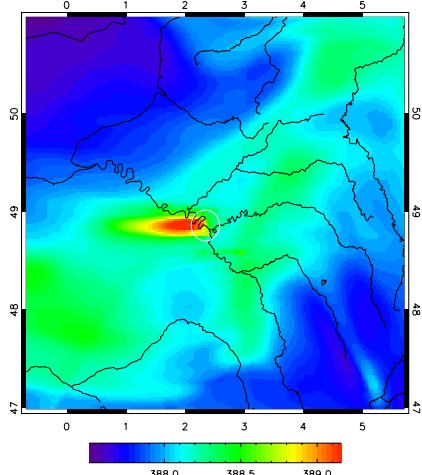

**Figure 1. Simulation of XCO$_2$ (in ppm) over the CHIMERE domain used in this study, on October 7$^{th}$ 2010 at 11:00 and at 2 km resolution using the operator described in section 2.5, the flux budgets given by Airparif and C-TESSEL and the model initial and boundary conditions given by the global LMDZ simulation. The Paris urban area approximately fits within the white circle located at the origin (East) of the plume in the middle of the domain. Most of the emissions of Paris are concentrated close to the centre of the urban area which corresponds to the administrative city of Paris.**



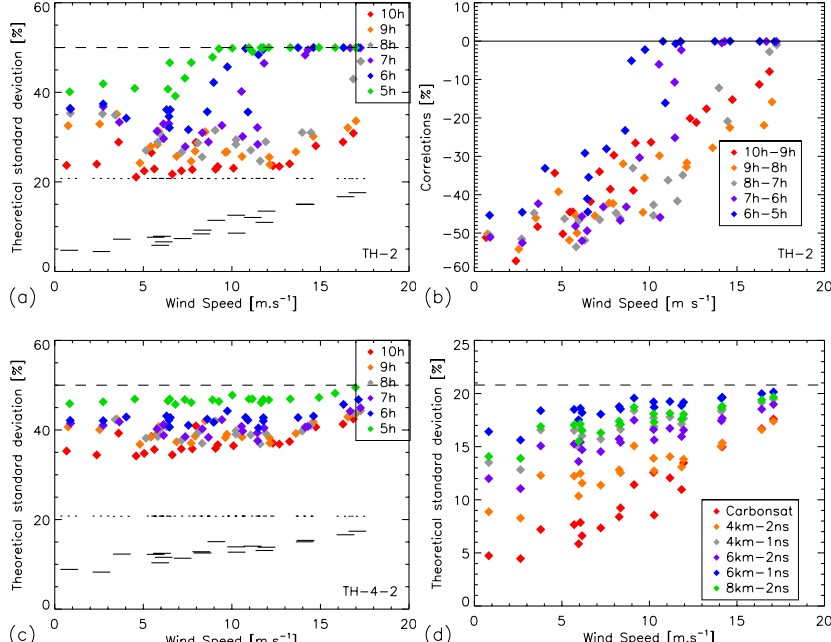

**Figure 2.** Prior and posterior uncertainties in the FF $CO_2$ emissions when controlling the hourly fluxes with the optimistic configurations of the OSSEs: results from the TH-2 (a,b) and TH-x-ns (c,d) experiments given as a function of the average wind speed over Paris at 700 magl (over the 6-hour duration of the simulation). Prior vs. posterior uncertainties in 1-hour mean (dashed black line vs. color dots) and 6-hour mean (dotted black segments vs. full black segments) emissions in TH-2 (a) and TH-4-2 (c). Correlations between prior (black lines) or posterior (color dots) uncertainties in 2 consecutive 1-hour mean emissions in TH-2 (b). Prior (black line) and posterior uncertainties (color dots) in 6-hour mean emissions in TH-2, TH-4-2, TH-4-1, TH-6-2, TH-6-1 and TH-8-2 (whose posterior uncertainties are given -in red, orange, light green, purple, blue, and dark green respectively) (d). Uncertainties are given in % of the emission budget given by Airparif. The colors of the dots are given as a function of the hour of the corresponding 1-hour emissions in (a,b,c) or as a function of the observation configuration in (d).



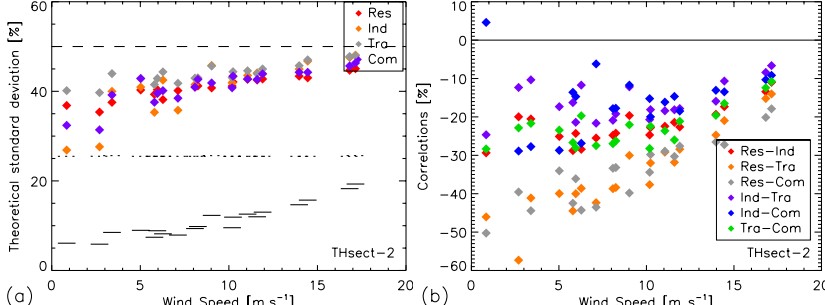

**Figure 3. Prior and posterior uncertainties in the FF $CO_2$ emissions when controlling the fluxes per type of process (Res=Residential combustion; Ind=industrial combustion; Tra=transportation; Com=commercial combustion) with the optimistic configurations of the OSSEs: results from the THsect-2 experiment given as a function of the average wind speed over Paris at 700 magl (over the 6-hour duration of the simulation). Prior vs. posterior uncertainties in sectoral (dashed black line vs. color dots) and total (dotted black segments vs. full black segments) 6-hour budgets of the emissions (a). Correlations between prior (black lines) or posterior (color dots) uncertainties in the emissions for two different sectors (b). The colors of the dots are given as a function of the hour of the corresponding 1-hour emissions.**



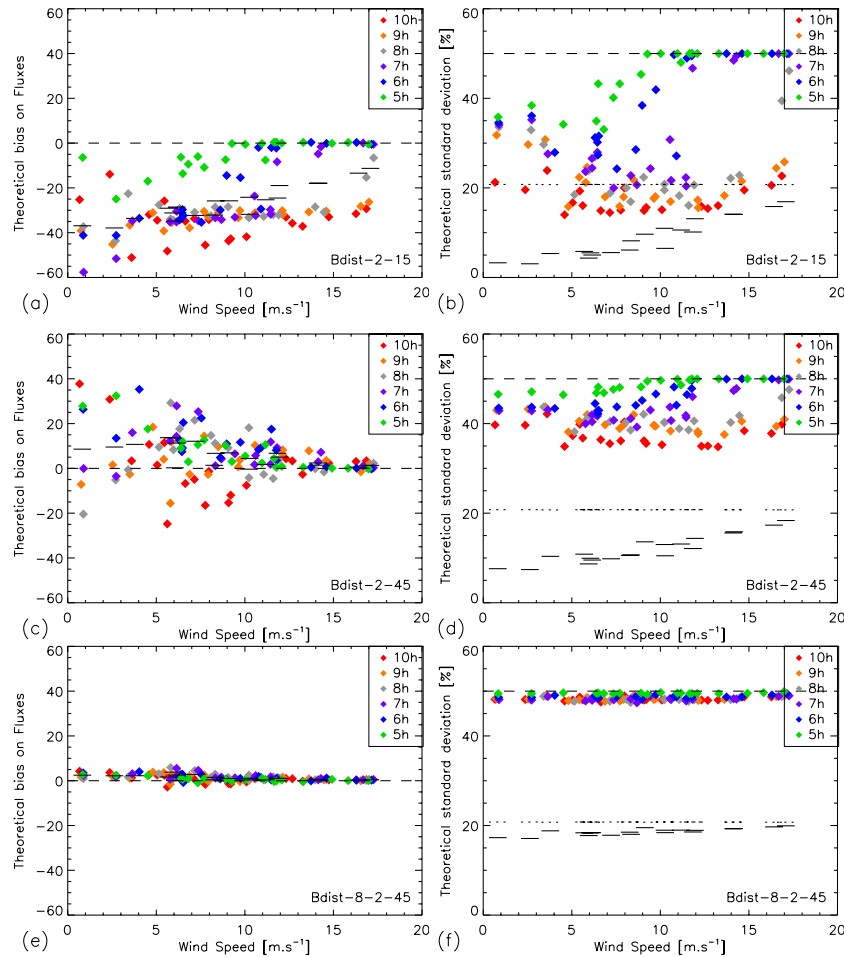

**Figure 4.** Prior and posterior biases and uncertainties in the FF $CO_2$ emissions when controlling the hourly fluxes with biases in the spatial distribution of the emissions: results from the Bdist-2-15 (a-b), Bdist-2-45 (c-d) and Bdist8-2-45 (e-f) experiments given as a function of the average wind speed over Paris at 700 magl (over the 6-hour duration of the simulation). Prior vs. posterior biases in 1-hour mean (dashed black line vs. color dots) and 6-hour mean (dashed black line vs. full black segments) emissions (a,c,e). Prior vs. posterior uncertainties in 1-hour mean (dashed black line vs. color dots) and 6-hour mean (dotted black segments vs. full black segments) emissions (b,d,f). The colors of the dots are given as a function of the hour of the corresponding 1-hour emissions.



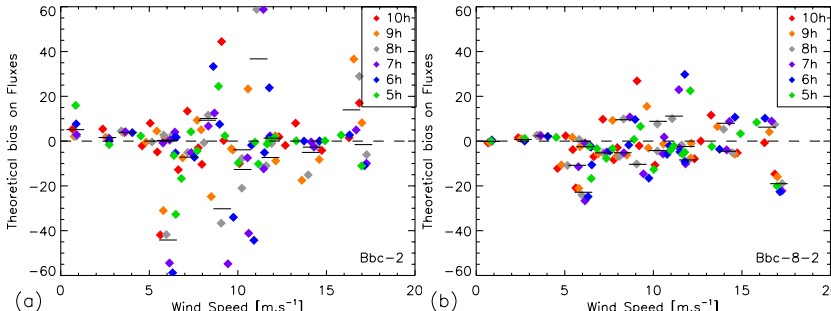

**Figure 5. Biases in the FF $CO_2$ emissions when controlling the hourly fluxes with biases in the boundary conditions: results from the Bbc-2 (a) and Bbc-8-2 (b) experiments given as a function of the average wind speed over Paris at 700 magl (over the 6-hour duration of the simulation). Prior vs. posterior biases in 1-hour mean (dashed black line vs. color dots) and 6-hour mean (dashed black line vs. full black segments) emissions. The colors of the dots are given as a function of the hour of the corresponding 1-hour emissions.**

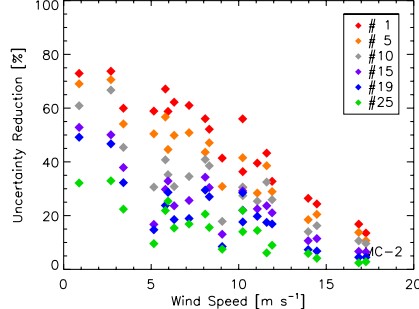

**Figure 6. Theoretical uncertainty reduction for the 6-hour mean emissions in the MC-2 experiments when using the 1st (red), 5th (orange), 10th (light green), 15th (purple), 19th (blue) and 25th (dark green) best observation sampling provided by the simulation of Buchwitz et al. (2013). The results are given as a function of the average wind speed over Paris at 700 magl (over the 6-hour duration of the simulation).**





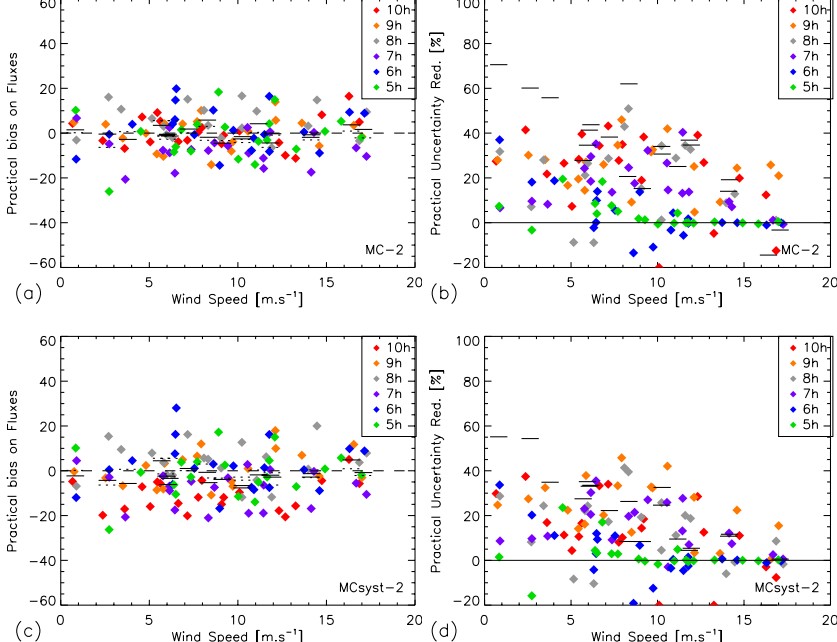

**Figure 7.** Prior and posterior biases and uncertainty reduction in the FF $CO_2$ emissions given by the statistics on the Monte Carlo ensembles of inverted emissions in the MC-2 (a,b) and MC-syst-2 (c,d) experiments with the simulation of the observation sampling and errors (including systematic errors in c,d) by Buchwitz et al. (2013). The results are given as a function of the average wind speed over Paris at 700 magl (over the 6-hour duration of the simulation). Posterior biases in 1-hour mean emissions (color dots) and prior vs posterior biases in 6-hour mean emissions (dotted black segments vs. full black segments) (a,c). Practical uncertainty reduction in 1-hour mean (color dots) and 6-hour mean (full black segments) emissions (b,d). The colors of the dots are given as a function of the hour of the corresponding 1-hour emissions.



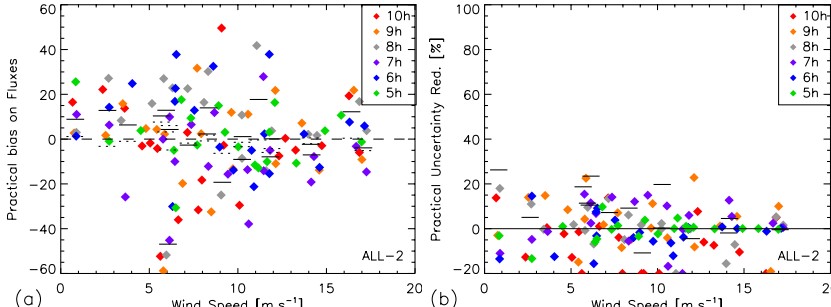

**Figure 8. Prior and posterior biases and uncertainty reduction in the FF CO$_2$ emissions given by the statistics on the Monte Carlo ensembles of inverted emissions in the ALL-2 experiment with the simulation of the observation sampling and errors (including systematic errors) by Buchwitz et al. (2013) and operator biases. Same legends as for the subfigures of figure 7.**

