# Peer review of "The potential of satellite spectro-imagery for monitoring CO2 emissions from large cities"

_Atmospheric Measurement Techniques, 2017_

## Referee Comment (RC1) · Anonymous Referee #3 · 25 Aug 2017

The authors report an observation simulation study of uncertainty reductions obtained using simulated CarbonSat observations of fossil fuel co2 emissions from the Paris, France urban area. In the absense of correlated (bias) errors, the authors estimate that a 50% reduction in uncertainty relative to the assumed prior uncertainty is obtained. The authors also include ă detailed estimates of the effect of bias errors.

General comments:

The problem is reasonably well conceived, the methods are sufficient to provide useful information. The results are useful in showing that use of CarbonSat data in regional CO2 inversions would reduce uncertainty in urban fossil fuel emissions. The paper would be useful contribution to AMT and could be published with revisions.

[Figure]

That said, the paper covers a lot of material and lacks clarity in some of the figures and description.

In particular, authors might add separate figures showing: 1) the predicted fossil and NEE signals within the modeling domain for one of the OSSE configurations 2) the pattern of sampling for each of the different OSSE configurations (TH-CS, TH-LS, SIM-CS from Table 1.)

The figures showing model results need better captions that describe that each panel shows points representing results for each of the 20 model days included in the OSSE (assuming I understand the figures correctly).

The assumption of relatively weak positive NEE for the October period doesn't allow consideration of other seasons. For example, strong NEE uptake in the growing season would result in potentially uncertain drawdown in XCO2 that could mask the fossil signal. This mentioned and potentially estimated numerically to provide a better sense of the seasonal cycle in CO2 sensing.

The methods description of systematic biases is so terse as to be unclear how large are the resulting signal (ppm) biases It would be helpful to have some additional figure or table to illustrate this before launching into results.

Last, I have not assessed the accuracy of the bias simulations so cannot comment on those results.

Specific comments:

abstract: The abstract should state the assumed prior model uncertainties (50% on all fluxes) before stating the uncertainty reductions provided by the observations.

page 7, Eq (1) & (2): The OSSE is estimating scaling factors for hourly (in a band of 6 hours) fossil fuel and NEE using maps of XCO2 covering the entire model domain. This effectively assumes that prior modeled fluxes do not contain significant correlated spatial errors. This should be identified as a limitation of the study and perhaps addressed

by comparing with modeled XCO2 signals obtained from a different prior model for emission (e.g., spatially uniform fluxes or a different flux models like EDGAR, GEIA, etc.).

page 11, line 23: What is the justification for assuming prior uncertainties for all fossil fuel and NEE sectors are equal to 50% ? This seems a rather weak constraint relative to uncertainties typically assumed for fossil fuel emissions, though perhaps not so far off for NEE.

Fig. 1: The figure axes or caption need to indicate units (degrees?)

[Figure]

---

## Referee Comment (RC2) · Anonymous Referee #4 · 18 Nov 2017

General comment.

The study is devoted to evaluation of using satellite observations for monitoring whole city anthropogenic CO2 emissions, focusing also on dependence of the emission estimation errors on different spatial resolution of satellite spectrometers, based on specifications of CarbonSat and Sentinel-5. After doing the OSSE with regional inverse modeling system based on 2 km resolution transport model, authors arrive at conclusion that high resolution (<4km) XCO2 imaging is preferable for this application. As the focus of the study is to evaluate different configurations of satellite observations, the topic fits to the subject area of AMT. The manuscript is well written, and doesn't require substantial editorial corrections. The paper can be accepted after addressing comments, requiring minor revision.

[Figure]

Detailed comments.

One real source of CO2 flux errors authors did not elaborate on is covariance between aerosol load and anthropogenic CO2. Aerosol load over large cities is leading to systematic biases in CO2 retrievals, the effect is being quantified in some studies (e.g. Jung et al., 2016).

Page 3, Lines 10-15 It would be worth adding a mention of recent results by Hakkarainen et al., (2016) and Nassar et al., (2017) obtained with OCO-2, and by Janardanan et al., (2016) with GOSAT. These studies are dealing with actual, not synthetic, data at relevant footprint resolution, therefore are providing hints on actual errors and biases in model and observations.

Page 9, Line 1 It is written as: "Consequently, there is no term associated with these emissions in the equations used in this study and they are ignored in the analysis of the results." To avoid confusing reader, it is better to give more detail on whether the anthropogenic fluxes outside of Paris are ignored completely or those are included in forward simulation, but not optimized.

Page 9, Line 28 Not everyone would agree with "This meteorological forcing does not account for urban land surface influences but we may neglect them for the OSSEs considered here". Breon et al., 2015 gave better excuse.

Page 32, Lines 3-5 Lack of available spatial detail is mentioned as common problem for many cities. There are two comments. One: This is said without going into detail of Airparif comparison to other high-resolution inventories like one used by Lauvaux et al., (2016), or produced by Tsagatakis et al., (2017). Second: for OSSE study, not having actual traffic count does not seem to be a major problem, synthetic traffic count should work.

References

Hakkarainen, J., Ialongo, I., and Tamminen, J.: Direct space-based observations of

anthropogenic CO2 emission areas from OCO-2, Geophysical Research Letters, 43, 11400-11406, 10.1002/2016GL070885, 2016.

Janardanan, R., Maksyutov, S., Oda, T., Saito, M., Kaiser, J., Ganshin, A., Stohl, A., Matsunaga, T., Yoshida, Y., and Yokota, T.: Comparing GOSAT observations of localized CO2 enhancements by large emitters with inventory-based estimates, Geophysical Research Letters, 43, 3486-3493, 10.1002/2016GL067843, 2016

Jung, Y., Kim, J., Kim, W., Boesch, H., Lee, H., Cho, C., and Goo, T.: Impact of Aerosol Property on the Accuracy of a CO2 Retrieval Algorithm from Satellite Remote Sensing, Remote Sensing, 8, 10.3390/rs8040322, 2016.

Lauvaux, T., Miles, N., Deng, A., Richardson, S., Cambaliza, M., Davis, K., Gaudet, B., Gurney, K., Huang, J., O'Keefe, D., Song, Y., Karion, A., Oda, T., Patarasuk, R., Razlivanov, I., Sarmiento, D., Shepson, P., Sweeney, C., Turnbull, J., and Wu, K.: High-resolution atmospheric inversion of urban CO2 emissions during the dormant season of the Indianapolis Flux Experiment (INFLUX), Journal of Geophysical Research-Atmospheres, 121, 5213-5236, 10.1002/2015JD024473, 2016.

Nassar, R., Hill, T., McLinden, C., Wunch, D., Jones, D., and Crisp, D.: Quantifying CO2 Emissions From Individual Power Plants From Space, Geophysical Research Letters, 44, 10045-10053, 10.1002/2017GL074702, 2017.

Tsagatakis, I., Brace, S., Passant, N., Pearson, B., Kiff, B., Richardson, J., and Ruddy, M.: UK Emission Mapping Methodology - A report of the National Atmospheric Emission Inventory 2015, Ricardo Energy & Environment, London, 1-63, 2017.

---

## Author Comment (AC1) · 16 Dec 2017

See a better presentation of the answers in the supplementary document.

Reviewer:

The authors report an observation simulation study of uncertainty reductions obtained using simulated CarbonSat observations of fossil fuel co2 emissions from the Paris, France urban area. In the absense of correlated (bias) errors, the authors estimate that a 50% reduction in uncertainty relative to the assumed prior uncertainty is obtained. The authors also include detailed estimates of the effect of bias errors. General comments: The problem is reasonably well conceived, the methods are sufficient to provide useful information. The results are useful in showing that use of CarbonSat

data in regional CO2 inversions would reduce uncertainty in urban fossil fuel emissions. The paper would be useful contribution to AMT and could be published with revisions.

Authors:

We thank the referee for this review and for his general assessment of the paper. Please find between his comments ("Reviewer") our answers and indications of how we improved the manuscript in line with them ("Authors").

Reviewer:

That said, the paper covers a lot of material and lacks clarity in some of the figures and description.

Authors:

Regarding the extent of the manuscript, we feel that a critical asset of our study flows from the various sensitivity tests that have been conducted, and from the series of messages that they bring individually or altogether. Therefore, we think that all the aspects and sensitivity tests that have been covered are important and that it was critical to analyze and discuss all of them in a single paper rather than to fragment them between different publications. We have improved or added some figures and legends according to the following comments from the referee.

Reviewer:

In particular, authors might add separate figures showing: 1) the predicted fossil and NEE signals within the modeling domain for one of the OSSE configurations

Authors:

An illustration of the response functions to hourly emissions from Paris and to hourly NEE in the modeling domain (see section 2.5.4) and of their aggregation into response functions for 6-hour emissions and NEE is now provided in the supplementary material

(as the new Figure S4). This figure is now referred to in section 2.5.4 and in section 3.1.

Reviewer:

2) the pattern of sampling for each of the different OSSE configurations (TH-CS, TH-LS, SIM- CS from Table 1.)

Authors:

The subfigures corresponding to TH-CS and TH-LS with and without perturbations associated with the measurement noise are now provided in the supplementary material (as the new Figure S1) and referred to in section 2.3. SIM-CS is illustrated in what used to be Figures S1 and S2 which are now Figures S2 and S3, and which have been referred to by section 2.3. We agree that such figures help understanding section 2.

Reviewer:

The figures showing model results need better captions that describe that each panel shows points representing results for each of the 20 model days included in the OSSE (assuming I understand the figures correctly).

Authors:

The reviewer understanding is correct, and this point has been clarified in the legends of the figures.

Reviewer:

The assumption of relatively weak positive NEE for the October period doesn't allow consideration of other seasons. For example, strong NEE uptake in the growing season would result in potentially uncertain drawdown in XCO2 that could mask the fossil signal. This mentioned and potentially estimated numerically to provide a better sense of the seasonal cycle in CO2 sensing.

Authors:

This is very difficult to quantify (even in terms of order of magnitude) without re-running the whole set of experiments with new NEE estimates. Our study shows some indices that the system does not fully separate the CO2 emission plume from the rest of the XCO2 scene, but it is difficult to anticipate how much this problem would impact the emission inversion if the amplitude of the NEE was much larger. We cannot extrapolate intuitively the low posterior correlations between uncertainties in the inverted NEE and uncertainties in the inverted emissions obtained in October to another month and then convert it into a level of error in the emission inversions due to problems of separations from the NEE for this other month. This topic was discussed in section 2.9, before the correlations between the posterior uncertainties in the NEE and the emissions were analyzed, and briefly reminded in section 5. We have now extended the corresponding text in section 5 to better address this specific point.

Reviewer:

The methods description of systematic biases is so terse as to be unclear how large are the resulting signal (ppm) biases. It would be helpful to have some additional figure or table to illustrate this before launching into results.

Authors:

What used to be Figure S1, which is now Figure S2, provides an extensive illustration of the values taken by the systematic errors and of their spatial patterns. We have now added a practical description of these errors in section 2.8.3.

Reviewer:

Last, I have not assessed the accuracy of the bias simulations so cannot comment on those results.

Authors:

These bias simulations have been generated and presented by Buchwitz et al. (2013). We can only refer to this publication for such a concern (which is regularly done in our paper). A few words have been added to say it more explicitly.

Reviewer:

Specific comments: abstract: The abstract should state the assumed prior model uncertainties (50% on all fluxes) before stating the uncertainty reductions provided by the observations.

Authors:

We have included this information.

Reviewer:

page 7, Eq (1) & (2): The OSSE is estimating scaling factors for hourly (in a band of 6 hours) fossil fuel and NEE using maps of XCO2 covering the entire model domain. This effectively assumes that prior modeled fluxes do not contain significant correlated spatial errors. This should be identified as a limitation of the study and perhaps addressed by comparing with modeled XCO2 signals obtained from a different prior model for emission (e.g., spatially uniform fluxes

Authors:

We do such an investigation with series of exp Bdist in sections 2.8.2, 4.1.1 and 4.3. The uncertainty in the spatial distribution of the emissions is also analyzed through the experiments TH-sect corresponding to section 3.3.

Reviewer:

or a different flux models like EDGAR, GEIA, etc.).

Authors:

This topic and the extent to which we have investigated it were already discussed in

section 5. We have now added some indications regarding the type of computations (the use of alternative inventories such as in Staufer et al. 2016) that could be conducted to refine such an investigation even though it was out of the scope of this paper to provide a precise assessment of the impact of all of sources of uncertainties.

Reviewer:

page 11, line 23: What is the justification for assuming prior uncertainties for all fossil fuel and NEE sectors are equal to 50% ? This seems a rather weak constraint relative to uncertainties typically assumed for fossil fuel emissions, though perhaps not so far off for NEE.

Authors:

This was discussed in section 5. There is very little information on the accuracy of city emission inventories at the hourly temporal scale. Most uncertainty estimates are made at the annual or monthly scales. The relative uncertainties for the hourly time scale could be much larger since the diurnal cycle is unknown. In addition, most inventories with temporal profiles are periodical with typical diurnal cycles for typical months, while weather patterns and specific events may have a large impact on the instantaneous emissions. Lastly, for cities other than Paris where there is a lack of information on the fossil fuel consumption, like in developing countries, the uncertainty may be much larger. This discussion has been extended to include NEE and further strengthen the justification for the fossil fuel emissions.

Reviewer:

Fig. 1: The figure axes or caption need to indicate units (degrees?)

Authors:

This has been done.

Please also note the supplement to this comment:

[Figure]

https://www.atmos-meas-tech-discuss.net/amt-2017-80/amt-2017-80-AC1-supplement.pdf

[Figure]

**Supplement:**

**Answers to the comments from anonymous Referee #3**

**Reviewer:**

The authors report an observation simulation study of uncertainty reductions obtained using simulated CarbonSat observations of fossil fuel co2 emissions from the Paris, France urban area. In the absense of correlated (bias) errors, the authors estimate that a 50% reduction in uncertainty relative to the assumed prior uncertainty is obtained. The authors also include detailed estimates of the effect of bias errors.

**General comments:**

The problem is reasonably well conceived, the methods are sufficient to provide useful information. The results are useful in showing that use of CarbonSat data in regional CO2 inversions would reduce uncertainty in urban fossil fuel emissions. The paper would be useful contribution to AMT and could be published with revisions.

**Authors:**

We thank the referee for this review and for his general assessment of the paper. Please find between his comments ("Reviewer") our answers and indications of how we improved the manuscript in line with them ("Authors").

**Reviewer:**

That said, the paper covers a lot of material and lacks clarity in some of the figures and description.

**Authors:**

Regarding the extent of the manuscript, we feel that a critical asset of our study flows from the various sensitivity tests that have been conducted, and from the series of messages that they bring individually or altogether. Therefore, we think that all the aspects and sensitivity tests that have been covered are important and that it was critical to analyze and discuss all of them in a single paper rather than to fragment them between different publications.

We have improved or added some figures and legends according to the following comments from the referee.

**Reviewer:**

In particular, authors might add separate figures showing: 1) the predicted fossil and NEE signals within the modeling domain for one of the OSSE configurations

**Authors:**

An illustration of the response functions to hourly emissions from Paris and to hourly NEE in the modeling domain (see section 2.5.4) and of their aggregation into response functions for 6-hour emissions and NEE is now provided in the supplementary material (as the new Figure S4). This figure is now referred to in section 2.5.4 and in section 3.1.

**Reviewer:**

2) the pattern of sampling for each of the different OSSE configurations (TH-CS, TH-LS, SIM- CS from Table 1.)

**Authors:**

The subfigures corresponding to TH-CS and TH-LS with and without perturbations associated with the measurement noise are now provided in the supplementary material (as the new Figure S1) and referred to in section 2.3.

SIM-CS is illustrated in what used to be Figures S1 and S2 which are now Figures S2 and S3, and which have been referred to by section 2.3.

We agree that such figures help understanding section 2.

**Reviewer:**

The figures showing model results need better captions that describe that each panel shows points representing results for each of the 20 model days included in the OSSE (assuming I understand the figures correctly).

**Authors:**

The reviewer understanding is correct, and this point has been clarified in the legends of the figures.

**Reviewer:**

The assumption of relatively weak positive NEE for the October period doesn't allow

consideration of other seasons. For example, strong NEE uptake in the growing season would result in potentially uncertain drawdown in XCO2 that could mask the fossil signal. This mentioned and potentially estimated numerically to provide a better sense of the seasonal cycle in CO2 sensing.

**Authors:**

This is very difficult to quantify (even in terms of order of magnitude) without re-running the whole set of experiments with new NEE estimates. Our study shows some indices that the system does not fully separate the CO2 emission plume from the rest of the XCO2 scene, but it is difficult to anticipate how much this problem would impact the emission inversion if the amplitude of the NEE was much larger. We cannot extrapolate intuitively the low posterior correlations between uncertainties in the inverted NEE and uncertainties in the inverted emission obtained in October to another month and then convert it into a level of error in the emission inversions due to problems of separations from the NEE for this other month.

This topic was discussed in section 2.9, before the correlations between the posterior uncertainties in the NEE and the emissions were analyzed, and briefly reminded in section 5. We have now extended the corresponding text in section 5 to better address this specific point.

**Reviewer:**

The methods description of systematic biases is so terse as to be unclear how large are the resulting signal (ppm) biases. It would be helpful to have some additional figure or table to illustrate this before launching into results.

**Authors:**

What used to be Figure S1, which is now Figure S2, provides an extensive illustration of the values taken by the systematic errors and of their spatial patterns. We have now added a practical description of these errors in section 2.8.3.

**Reviewer:**

Last, I have not assessed the accuracy of the bias simulations so cannot comment on those results.

**Authors:**

These bias simulations have been generated and presented by Buchwitz et al. (2013). We can only refer to this publication for such a concern (which is regularly done in our paper). A few

words have been added to say it more explicitly.

**Reviewer:**

Specific comments:

abstract: The abstract should state the assumed prior model uncertainties (50% on all fluxes) before stating the uncertainty reductions provided by the observations.

**Authors:**

We have included this information.

**Reviewer:**

page 7, Eq (1) & (2): The OSSE is estimating scaling factors for hourly (in a band of 6 hours) fossil fuel and NEE using maps of XCO2 covering the entire model domain. This effectively assumes that prior modeled fluxes do not contain significant correlated spatial errors. This should be identified as a limitation of the study and perhaps addressed by comparing with modeled XCO2 signals obtained from a different prior model for emission (e.g., spatially uniform fluxes

**Authors:**

We do such an investigation with series of exp Bdist in sections 2.8.2, 4.1.1 and 4.3. The uncertainty in the spatial distribution of the emissions is also analyzed through the experiments TH-sect corresponding to section 3.3.

**Reviewer:**

or a different flux models like EDGAR, GEIA, etc.).

**Authors:**

This topic and the extent to which we have investigated it were already discussed in section 5. We have now added some indications regarding the type of computations (the use of alternative inventories such as in Staufer et al. 2016) that could be conducted to refine such an investigation even though it was out of the scope of this paper to provide a precise assessment of the impact of all of sources of uncertainties.

**Reviewer:**

page 11, line 23: What is the justification for assuming prior uncertainties for all fossil fuel and

NEE sectors are equal to 50% ? This seems a rather weak constraint relative to uncertainties typically assumed for fossil fuel emissions, though perhaps not so far off for NEE.

**Authors:**

This was discussed in section 5. There is very little information on the accuracy of city emission inventories at the hourly temporal scale. Most uncertainty estimates are made at the annual or monthly scales. The relative uncertainties for the hourly time scale could be much larger since the diurnal cycle is unknown. In addition, most inventories with temporal profiles are periodical with typical diurnal cycles for typical months, while weather patterns and specific events may have a large impact on the instantaneous emissions. Lastly, for cities other than Paris where there is a lack of information on the fossil fuel consumption, like in developing countries, the uncertainty may be much larger. This discussion has been extended to include NEE and further strengthen the justification for the fossil fuel emissions.

**Reviewer:**

Fig. 1: The figure axes or caption need to indicate units (degrees?)

Authors:

This has been done.

**The potential of satellite spectro-imagery for monitoring CO2 emissions from large cities**

Grégoire Broquet1, François-Marie Bréon1, Emmanuel Renault1,\*, Michael Buchwitz2, Maximilian Reuter2, Heinrich Bovensmann2, Frédéric Chevallier1, Lin Wu1, Philippe Ciais1

[revised manuscript text omitted]

Therefore, but we may neglect them these parameters for the our OSSEs considered here.

The CO2 mixing ratios at the lateral and top boundaries of the CHIMERE model and the initial conditions at 5:00 (i.e.  $c^{fixed}$ ) are imposed using outputs from a global LMDZ simulation at 3.75° (longitude) × 2.5° (latitude) resolution assimilating in situ CO2 data from a global ground based measurement network (Chevallier et al., 2010).

**2.5.3 Vertical integration of the CO2 column**

15 The third operator is the computation of XCO2 data **y** for a given satellite observation sampling based on the 3D CO2 concentrations at 11:00 in the CHIMERE domain, from the surface up to Ptop: **c** → **y** = **M**integ **c**. Part of this computation includes the aggregation of the 2 km horizontal resolution CO2 fields at the chosen satellite image resolution. For the sake of simplicity and since we use synthetic data only, it is assumed that the satellite observation has a uniform vertical weighting function for each horizontal pixel of a given satellite space sampling (see above section 2.3). For a given horizontal pixel at 20 latitude *lat* and longitude *lon*, XCO2 is thus computed from the average of the vertically distributed CO2 mole fractions:

$$XCO_2(lon, lat) = \frac{1}{P_{surf}(lon, lat)} \left( \int_{P_{top}}^{P_{surf}(lon, lat)} CO_2(lon, lat, p) dp + \overline{CO_2} \left( P_{top} \right) P_{top} \right)$$
(3)

where p is the pressure and  $P_{surf}$  is the surface pressure. In this equation, a uniform concentration equal to the horizontal average of the top level mixing ratios in CHIMERE:  $\overline{CO_2}(P_{top})$  is assumed to apply at all pressures lower than  $P_{top}$ . This simple approximation is used since the surface fluxes in the simulation domain do not impact the concentration close to the

25

simple approximation is used since the surface fluxes in the simulation domain do not impact the concentration close to the top of the model within the simulation time. It ignores the signature of remote fluxes above 500 hPa and we implicitly assume that this signature will not significantly impact the inversion of the emissions from the Paris area. This equation also ignores the slant path of the columns that would be measured by a satellite within the model. Finally it ignores the potential impact of the water vapour content of the atmosphere on the vertical weighting function of the satellite measurements.

**2.5.4 Practical computation of the observation operator**

 $\mathbf{s} \rightarrow \mathbf{y} = \mathbf{M} \mathbf{s} + \mathbf{y}^{\text{fixed}}$ The observation operator can be rewritten  $\mathbf{s} \rightarrow \mathbf{y} = \mathbf{M}_{\text{integ}} \mathbf{M}_{\text{transport}} \mathbf{M}_{\text{inventory}} \mathbf{\lambda} + b [1, 1 \dots 1]_{n_y}^T + \mathbf{y}^{\text{fixed}}$  where  $[1, 1 \dots 1]_{n_y}$  has the length  $n_y$  corresponding to the 1D representation of the observation space. In order to apply the inversion equations analytically (see section 2.6 below), the **M** matrix is built explicitly. For this, the  $\mathbf{M}_{integ} \mathbf{M}_{transport} \mathbf{M}_{inventory}$  matrix is built by computing each of its column  $\mathbf{m}_{j}$ through the application of the operator  $\lambda \rightarrow y = M_{\text{integ}} M_{\text{transport}} M_{\text{inventory}} \lambda$  to unit vectors  $\lambda_j = [0 \dots 0 \ 1 \ 0 \dots 0]^T$  where only the jth control parameter corresponding to a flux scaling factor is set to 1 and others are set to 0. The small number of control parameters makes the number of simulations  $\lambda_j \rightarrow m_j = M_{\text{integ}} M_{\text{transport}} M_{\text{inventory}} \lambda_j$  computationally affordable. The columns  $\mathbf{m}_{i}$  correspond to the response functions in terms of XCO2 to each of the Airparif or C-TESSEL flux component controlled by a scaling factor (see an illustration of these response functions in figure S4). The full observation operator matrix **M** is obtained by adding the column  $[1,1...1]_{n_v}^T$  to  $\mathbf{M}_{\text{integ}} \mathbf{M}_{\text{transport}} \mathbf{M}_{\text{inventory}}$  which corresponds to the homogeneous distribution in space of b the background XCO2 at 11:00.

**2.6 Theoretical framework of the atmospheric inversion**

The theoretical framework of the inversion system used here in the OSSEs is the one traditionally used for atmospheric 15 inversions. It is based on the Bayesian update of a statistical prior estimate of the control vector **s**, using statistical 16 information from the assimilation of the measurements  $\mathbf{y}^{\circ}$  in the observation operator. The usual assumption is that the prior 17 estimate has a Gaussian distribution  $N(\mathbf{s}^{\mathsf{b}}, \mathbf{B})$  and that the distribution of the misfits between the simulated observations 18  $\mathbf{M} \mathbf{s} + \mathbf{y}^{\mathsf{fixed}}$  and  $\mathbf{y}^{\circ}$  that are not due to errors in  $\mathbf{s}$ , i.e. the so-called observation errors, which include atmospheric transport 18 and representation errors, is unbiased, has a Gaussian distribution  $N(\mathbf{0}, \mathbf{R})$ , and is not correlated with the prior uncertainty. In

20 that case, the Bayesian update of the estimate of s, called hereafter the posterior distribution, is a Gaussian distribution N(sa,A) with sa being the best estimate of the actual s knowing sb and yo, and A characterizing the uncertainty in this estimate. The problem simplifies into deriving (Tarantola, 2005):

$$\mathbf{A} = [\mathbf{B}^{-1} + \mathbf{M}^{\mathrm{T}} \mathbf{R}^{-1} \mathbf{M}]^{-1}$$
(4)

and

5

10

$$\mathbf{s}^{\mathbf{a}} = \mathbf{s}^{\mathbf{b}} + \mathbf{K} \left( \mathbf{y}^{\mathbf{o}} - \mathbf{M} \, \mathbf{s}^{\mathbf{
[revised manuscript text omitted]

---

## Author Comment (AC2) · 16 Dec 2017

Please find a clearer presentation of the answers in the supplementary document.

Reviewer:

General comment. The study is devoted to evaluation of using satellite observations for monitoring whole city anthropogenic CO2 emissions, focusing also on dependence of the emission estimation errors on different spatial resolution of satellite spectrometers, based on specifications of CarbonSat and Sentinel-5. After doing the OSSE with regional inverse modeling system based on 2 km resolution transport model, authors arrive at conclusion that high resolution (<4km) XCO2 imaging is preferable for this application. As the focus of the study is to evaluate different configurations of satellite

observations, the topic fits to the subject area of AMT. The manuscript is well written, and doesn't require substantial editorial corrections. The paper can be accepted after addressing comments, requiring minor revision.

Authors:

We thank the reviewer for this general evaluation of our paper and for his useful comments. Please find between these comments ("Reviewer") our answers and indications of how we improved the manuscript in line with them ("Authors").

Reviewer:

Detailed comments. One real source of $CO_2$ flux errors authors did not elaborate on is covariance between aerosol load and anthropogenic $CO_2$. Aerosol load over large cities is leading to systematic biases in $CO_2$ retrievals, the effect is being quantified in some studies (e.g. Jung et al., 2016).

Authors:

This is now mentioned in section 5.

Reviewer:

Page 3, Lines 10-15 It would be worth adding a mention of recent results by Hakkarainen et al., (2016) and Nassar et al., (2017) obtained with OCO-2, and by Janardanan et al., (2016) with GOSAT. These studies are dealing with actual, not synthetic, data at relevant footprint resolution, therefore are providing hints on actual errors and biases in model and observations.

Authors:

These three publications are cited in the revised version of the manuscript.

Reviewer:

Page 9, Line 1 It is written as: "Consequently, there is no term associated with these

emissions in the equations used in this study and they are ignored in the analysis of the results." To avoid confusing reader, it is better to give more detail on whether the anthropogenic fluxes outside of Paris are ignored completely or those are included in forward simulation, but not optimized.

Authors:

We do not really have to consider a "forward simulation" in this study since we only solve for equations 4, 7, 11 and 12 but not for equation 5. We just had to consider it as a matter of illustration when producing Figures 1 and similar ones in the supplementary material, where these emissions outside the Paris area are ignored. This point is now clarified in section 2.5.1.

Reviewer:

Page 9, Line 28 Not everyone would agree with "This meteorological forcing does not account for urban land surface influences but we may neglect them for the OSSEs considered here". Breon et al., 2015 gave better excuse.

Authors:

Breon et al., 2015 considered ground based stations within and at the edge of the Paris urban area. This is very different from considering satellite observations, which focus on a whole plume whose length is more than 100 km downwind of the urban area. The other critical difference is that Breon et al. 2015 dealt with real data and thus needed to catch actual patterns of the transport rather than just to produce "realistic" simulations. The situation is the opposite for this paper. We have extended this piece of text to clarify it.

Reviewer:

Page 32, Lines 3-5 Lack of available spatial detail is mentioned as common problem for many cities. There are two comments. One: This is said without going into detail of Airparif comparison to other high-resolution inventories like one used by Lauvaux

et al., (2016), or produced by Tsagatakis et al., (2017). Second: for OSSE study, not having actual traffic count does not seem to be a major problem, synthetic traffic count should work.

Authors:

The sentence was misleading and has been improved. We did not aim at comparing Airparif to inventories for other cities. We meant that, to our knowledge: 1) Airparif provides a state-of-the-art quantitative description of the emissions from the Paris urban area at high spatial (1 km) and temporal (hourly) resolution 2) the current existing inventories with temporal variations (including the Airparif inventory) describe relatively homogeneous and cycling temporal variations of the emissions even when they have been derived from precise data at a high spatial and temporal resolution. The UK-NAEI inventory presented by Tsagatakis et al., (2017) has no temporal variation. The temporal variations in the Hestia inventory used by Lauvaux et al. (2016) are based on average diurnal and weekly cycles. The point was not really about the need for precise (actual) data on e.g. traffic count, but about using realistic hourly variations of the emissions for each 2 km grid cell of the transport model rather than homogeneous and cycling temporal variations in the inventory (either from real or synthetic data) to avoid having a too "diffuse" representation of the emissions.

Reviewer:

References
 Hakkarainen, J., Ialongo, I., and Tamminen, J.: Direct space-based observations of anthropogenic CO2 emission areas from OCO-2, Geophysical Research Letters, 43, 11400-11406, 10.1002/2016GL070885, 2016. Janardanan, R., Maksyutov, S., Oda, T., Saito, M., Kaiser, J., Ganshin, A., Stohl, A., Matsunaga, T., Yoshida, Y., and Yokota, T.: Comparing GOSAT observations of localized CO2 enhancements by large emitters with inventory-based estimates, Geophysical Research Letters, 43, 3486-3493, 10.1002/2016GL067843, 2016 Jung, Y., Kim, J., Kim, W., Boesch, H., Lee, H., Cho, C., and Goo, T.: Impact of Aerosol Property on

the Accuracy of a CO2 Retrieval Algorithm from Satellite Remote Sensing, Remote Sensing, 8, 10.3390/rs8040322, 2016. Lauvaux, T., Miles, N., Deng, A., Richardson, S., Cambaliza, M., Davis, K., Gaudet, B., Gurney, K., Huang, J., O'Keefe, D., Song, Y., Karion, A., Oda, T., Patarasuk, R., Razlivanov, I., Sarmiento, D., Shepson, P., Sweeney, C., Turnbull, J., and Wu, K.: High-resolution atmospheric inversion of urban CO2 emissions during the dormant season of the Indianapolis Flux Experiment (INFLUX), Journal of Geophysical Research-Atmospheres, 121, 5213-5236, 10.1002/2015JD024473, 2016. Nassar, R., Hill, T., McLinden, C., Wunch, D., Jones, D., and Crisp, D.: Quantifying CO2 Emissions From Individual Power Plants From Space, Geophysical Research Letters, 44, 10045-10053, 10.1002/2017GL074702, 2017. Tsagatakis, I., Brace, S., Passant, N., Pearson, B., Kiff, B., Richardson, J., and Ruddy, M.: UK Emission Mapping Methodology - A report of the National Atmospheric Emission Inventory 2015, Ricardo Energy & Environment, London, 1-63, 2017.

Please also note the supplement to this comment:
https://www.atmos-meas-tech-discuss.net/amt-2017-80/amt-2017-80-AC2-supplement.pdf

[Figure]

**Supplement:**

**Answers to the comments from anonymous Referee #4**

Reviewer:

General comment.

The study is devoted to evaluation of using satellite observations for monitoring whole city anthropogenic CO2 emissions, focusing also on dependence of the emission estimation errors on different spatial resolution of satellite spectrometers, based on specifications of CarbonSat and Sentinel-5. After doing the OSSE with regional inverse modeling system based on 2 km resolution transport model, authors arrive at conclusion that high resolution (<4km) XCO2 imaging is preferable for this application. As the focus of the study is to evaluate different configurations of satellite observations, the topic fits to the subject area of AMT. The manuscript is well written, and doesn't require substantial editorial corrections. The paper can be accepted after addressing comments, requiring minor revision.

Authors:

We thank the reviewer for this general evaluation of our paper and for his useful comments. Please find between these comments ("Reviewer") our answers and indications of how we improved the manuscript in line with them ("Authors").

Reviewer:

Detailed comments.

One real source of CO2 flux errors authors did not elaborate on is covariance between aerosol load and anthropogenic CO2. Aerosol load over large cities is leading to systematic biases in CO2 retrievals, the effect is being quantified in some studies (e.g. Jung et al., 2016).

Authors:

This is now mentioned in section 5.

Reviewer:

Page 3, Lines 10-15 It would be worth adding a mention of recent results by Hakkarainen et al., (2016) and Nassar et al., (2017) obtained with OCO-2, and by Janardanan et al., (2016) with

GOSAT. These studies are dealing with actual, not synthetic, data at relevant footprint resolution, therefore are providing hints on actual errors and biases in model and observations.

Authors:

These three publications are cited in the revised version of the manuscript.

Reviewer:

Page 9, Line 1 It is written as: "Consequently, there is no term associated with these emissions in the equations used in this study and they are ignored in the analysis of the results." To avoid confusing reader, it is better to give more detail on whether the anthropogenic fluxes outside of Paris are ignored completely or those are included in forward simulation, but not optimized.

Authors:

We do not really have to consider a "forward simulation" in this study since we only solve for equations 4, 7, 11 and 12 but not for equation 5. We just had to consider it as a matter of illustration when producing Figures 1 and similar ones in the supplementary material, where these emissions outside the Paris area are ignored. This point is now clarified in section 2.5.1.

Reviewer:

Page 9, Line 28 Not everyone would agree with "This meteorological forcing does not account for urban land surface influences but we may neglect them for the OSSEs considered here". Breon et al., 2015 gave better excuse.

Authors:

Breon et al., 2015 considered ground based stations within and at the edge of the Paris urban area. This is very different from considering satellite observations, which focus on a whole plume whose length is more than 100 km downwind of the urban area. The other critical difference is that Breon et al. 2015 dealt with real data and thus needed to catch actual patterns of the transport rather than just to produce "realistic" simulations. The situation is the opposite for this paper. We have extended this piece of text to clarify it.

Reviewer:

Page 32, Lines 3-5 Lack of available spatial detail is mentioned as common problem for many cities. There are two comments. One: This is said without going into detail of Airparif comparison to other high-resolution inventories like one used by Lauvaux et al., (2016), or

produced by Tsagatakis et al., (2017). Second: for OSSE study, not having actual traffic count does not seem to be a major problem, synthetic traffic count should work.

Authors:

The sentence was misleading and has been improved. We did not aim at comparing Airparif to inventories for other cities. We meant that, to our knowledge:

1) Airparif provides a state-of-the-art quantitative description of the emissions from the Paris urban area at high spatial (1 km) and temporal (hourly) resolution

2) the current existing inventories with temporal variations (including the Airparif inventory) describe relatively homogeneous and cycling temporal variations of the emissions even when they have been derived from precise data at a high spatial and temporal resolution. The UK-NAEI inventory presented by Tsagatakis et al., (2017) has no temporal variation. The temporal variations in the Hestia inventory used by Lauvaux et al. (2016) are based on average diurnal and weekly cycles.

The point was not really about the need for precise (actual) data on e.g. traffic count, but about using realistic hourly variations of the emissions for each 2 km grid cell of the transport model rather than homogeneous and cycling temporal variations in the inventory (either from real or synthetic data) to avoid having a too "diffuse" representation of the emissions.

Reviewer:

[revised manuscript text omitted]

(a) Response function for 7−8AM emissions

(b) Response function for 5−11AM emissions

(c) Response function for 7−8AM NEE

(d) Response function for 5−11AM NEE

**Figure S4. Simulation of the XCO$_2$ response functions of different flux components (in ppm) as seen from space over the CHIMERE domain used in this study, on October 7$^{th}$ 2010 at 11:00 and at 2 km resolution using the operator described in section 2.5, the computations described in section 2.5.4, and the flux budgets given by Airparif or C-TESSEL. a) Response function for the emissions from Paris between 7:00 and 8:00. a) Response function for the emissions from Paris between 5:00 and 11:00 (i.e., sum of the response functions for the hourly emissions from Paris between 5:00 and 11:00). c) Response function for the NEE between 7:00 and 8:00. d) Response function for the NEE between 5:00 and 11:00 (i.e., sum of the response functions for the hourly NEE between 5:00 and 11:00). The longitudes and latitudes of the domain are indicated in degrees East and North.**